# Secondary teachers' competencies and attitude: A mediated multigroup model based on usefulness and enjoyment to examine the differences between key dimensions of STEM teaching practice

**Fabiola Sanda Chiriacescu**[1], **Bogdan Chiriacescu**[1], **Alina Elena Grecu**[2], **Cristina Miron**[1], **Ion Ovidiu Panisoara**[2], **Iuliana Mihaela Lazar**[2]*

**1** Faculty of Physics, University of Bucharest, Bucharest, Romania, **2** Faculty of Psychology and Educational Sciences, University of Bucharest, Bucharest, Romania

* iulia.lazar@fpse.unibuc.ro

## Abstract

This research explores the mediating role of perceived usefulness and enjoyment of science, technology, engineering, and mathematics (STEM) teaching practice between secondary teachers' competencies and attitudes in the formal educational context. Also, the research aimed to examine if the relationships between model constructs differ by STEM teaching practice dimension (e.g., Inquiry-based learning (IBL) and Integration of STEM content (INT)). We synthesized the will, skill, tool model (WST), technology acceptance model (TAM) and flow theory (FLT) to develop a theoretical model predicting teacher attitude under the influence of Competencies, Perceived Usefulness and Perceived Enjoyment. Therefore, a mediated multigroup model with validated data from three hundred Romanian secondary teachers who completed questionnaires related to their competencies, perceived usefulness, enjoyment, and attitude toward STEM teaching practices was used. Two comparative teacher survey studies were carried out: one for IBL and one for INT. There are direct and positive relationships between Competencies and Attitude, Competencies and Enjoyment, Competencies and Usefulness, and Enjoyment and Attitude for both IBL and INT teaching practices. The partial least squares path modeling (PLS-SEM) results showed that the control variables had no significant impact on attitude. This research supports evidence for the belief that teachers' competency is a key predictor of attitude. Precisely, the positive strong direct effect of Competencies on Attitude is similar for IBL (β = 0.49, t = 7.46, p < 0.001; $f^2$ (Effect size) = 0.29) with for INT teaching practice (β = 0.46, t = 6.46, p < 0.001; $f^2$ (Effect size) = 0.22). Interestingly, this research showed that the perceived Usefulness and Enjoyment partially mediated the association between Competencies and Attitude in both case studies. Understanding the mediating role of perceived usefulness and enjoyment for each STEM practice would help teachers successfully implement STEM education.

**Data Availability Statement:** All relevant data are within the manuscript and its Supporting Information files.

**Funding:** The author(s) received no specific funding for this work.

**Competing interests:** The authors have declared that no competing interests exist.

## Introduction

The importance of the science, technology, mathematics, and engineering (STEM) concept for education is unanimously recognized by researchers [1, 2], but its definition is still unclear. According to Zizka (2021) [3, p. 2], integrated STEM education is a "*technique to offer learners the theoretic knowledge and technical-related competencies*". From this perspective, STEM education has become a tactical option for education reform due to its potential role in future work skills [4].

The integrated STEM curriculum has developed from the desire to fulfil the need to understand phenomena and events from reality [5]. The usefulness and enjoyment of learning in a multidisciplinary context using STEM teaching practice [6] are necessary for encouraging a positive attitude towards education. Furthermore, the level of teacher competence resulting from interdisciplinarity, research and collaboration is higher than that achieved at the disciplinary level without interactive and innovative elements [7]. Consequently, most STEM definitions consist of an interdisciplinary view of teaching based on real-world lessons, considering the learning context [8].

An integrated STEM curriculum facilitated collaboration between students and between students and teachers [5] and stimulated learners' interest and motivation [9]. Students come into direct contact with the surrounding reality, with concrete real-life situations, which allow them to solve practical problems of existence [10]. How the students' competencies meet their subsequent social needs represents a significant challenge for officials. Developing a positive attitude in teachers to adopt STEM teaching practices is necessary. Moreover, the enhancement of teachers' curiosity of knowledge for exploration and discovery and their desire to experience new things has represented a continuing concern of educational policies [11, 12].

The generalized conception of IBL [13] reflects an educational strategy that promotes active participation and empowerment of students [14], in which they solve problems using research skills [15]. Since IBL involves a paradigm shift, the responsibility for learning falls mainly on learners. The multifaceted effort of IBL will increase students' engagement in STEM activities [16]. Nevertheless, the IBL practice has been demonstrated to be a "useful strategy for improving students' motivation in STEM subjects" as Silm and collaborators (2017) stated [17]. By increasing the engagement and motivation of students, it is possible to help the efficiency of the learning process. Therefore, the educational climate can be improved. Exploring the factors affecting learners' attitudes is essential to fulfilling this aspiration.

Investigating the dimensions that influence secondary teachers' attitudes has been the subject of numerous studies, especially from the need to identify as precisely as possible the constructs that positively impact the attitude [18] to facilitate learning activities essential for students' knowledge. The interaction between attitudes toward a field (will), skills in that field (skill), and access to technology (tool) [19] was frequently evaluated with the will, skill, tool (WST) model developed by Knezek and Christensen [20]. The integrated theoretical framework was completed by constructs from technology acceptance model and flow theory utilized in a digital environment. In addition, there are suggestions that the absence of a negative attitude is critical [21] for the successful adoption of educational teaching practice. Nevertheless, developing students' 21st Century Competencies [22] is particularly challenging for teachers.

The results of previous studies showed that numerous predictors, e.g., competence, perception of effectiveness, and ease of use of technology or teacher experience, were investigated as attitude antecedents [19]. Perceived enjoyment is a powerful predictor of attitude [23]. Perceived usefulness affected the attitude [24]. The Inquiry-based learning (IBL) practice has been demonstrated to be efficient in increasing students' motivation in STEM subjects [17]. The quality of the IBL has been associated with the degree of understanding of experimental

activities [25]. More teachers advocate integrated teaching and learning based on interdisciplinary and investigative instructional strategies. The investigative approach has changed the emphasis of instruction from memorizing concepts to investigative learning. In such contexts, the development of the critical thinking skills of students will be assured because they are forced to find answers to various problems [26].

Nevertheless, improving students' competencies using technology as a bridge to combine multiple disciplines has been a significant challenge for several decades [27]. This statement is because STEM integration requires interdisciplinary collaboration, and teachers are usually qualified to teach specific knowledge of a topic [7]. Real-world problems, which are STEM subjects, are frequently interdisciplinary and arise in complex systems [28]. Natural sciences are often treated as separate disciplines [29], and students were not encouraged to approach learning content by making connections between different subjects. The preparation and implementation of IBL and INT require a great deal of effort on the part of teachers. Unfortunately, material resources are often lacking.

Despite all the obstacles, some teachers believe in the benefits of STEM education, where discipline-specific content is not divided [28, 30]. They addressed and treated STEM subjects as one dynamic like Integration of STEM content (INT) practice, as Merrill (2009) stated [30, pg.1]. This approach will improve problem-solving skills and develop students' critical and analytical thinking [31]. However, how teachers involve students in significant inquiry activities continues to be explored [32]. It can be concluded that STEM practice is a nonlinear method [33] that is not well understood [24].

The distance education imposed by the COVID-19 pandemic [34] has accelerated the adoption of instructional strategies based on online digital tools into the educational process, facilitating technology acceptance by all teachers. However, not all instructional strategies that promote student learning in exploratory and transdisciplinary ways, like SEM practices, are accepted on a large scale. According to Thibaut and collaborators [9], various distinctive but connected key facets of integrated STEM were observed. Also, features of integrated STEM education are essential since they create the required settings for the teaching procedures. These aspects were convincing to perform this research based on a modified WST model.

Two new dimensions (i.e., perceived usefulness and enjoyment) of STEM instructional strategies have been proposed alongside skill and will. These two features mediate between skill and will and replace the tool construct of the WST model. The present research was designed to explore the possible mediating effect of perceived usefulness and enjoyment between competencies and attitudes across secondary teachers.

Until now, limited attention has been paid to highlighting differences and similarities between IBL and INT teaching practices. The relationship between competencies and attitude to use IBL or INT teaching practices is still unclear. Understanding how teachers' competencies influence the acceptance of the STEM curriculum is essential. Briefly, two of the key facets of integrated STEM (e.g., *inquiry-based learning (IBL) and integration of STEM content (INT)*) were subjects of the present research. This study used a specially designed model to fill the gaps found.

The research questions (RQ) that guided the research are as follows:

1. Do competencies directly affect the attitude of secondary teachers who adopted IBL and INT as teaching practices?

2. Do perceived usefulness and enjoyment mediate the relationship between competencies and attitudes of secondary teachers who adopted IBL and INT as teaching practices?

3. Does the strength of the relationship between competencies and attitude mediated by enjoyment and usefulness differ between IBL and INT teaching practices?

## Literature review and hypothesis summary

**Independent variable.** *Competencies*. Many teachers are not "natives" of digital technologies, as their students are. Teachers must adapt to the latest educational technologies acquiring skills to use them. The expansion of the Internet network and new technologies deeply affected the scientific accuracy of the available online content. For these reasons, teachers must develop new skills related to synthesizing a large amount of information, including interdisciplinary areas [21], and extracting essential and accurate data from them. The necessity to adopt and extend critical thinking, problem-solving, communication, collaboration, creativity, and innovation influences teachers' competencies [35].

These challenges can only be solved by identifying new, creative teaching practices. The third component of competencies is identifying values. Students act correctly and beyond the classroom, including exercising life, work, ethics, emotions, or cultural competencies in the online environment. For the training and development of STEM competence of students, it is necessary that teachers first possess these competencies [36]. These demands include theoretical knowledge in science and mathematical competence, understanding of investigative skills, and competence to design, facilitate and evaluate a scientific research activity.

A capable teacher trusts himself and effectively meets objectives and tasks, addressing challenges [37]. Students perceived teachers who used integrative and transdisciplinary strategies as more efficient [38]. Adopting STEM strategies also stimulated their cognitive involvement and influenced their involvement and perseverance in school activities. Therefore, it is natural to expect teachers' competencies regarding STEM teaching practices to influence their attitudes. Involving teachers in an effective but also attractive course requires training and documentation. The teacher has responsibility [39] for the achievement of new competencies.

In this study, the external variable assessed teachers' competencies to identify subjects suitable for teaching through STEM teaching practices. Briefly, this dimension measured teachers' beliefs about the roles of STEM teaching mode in students' understanding of the subject matter led. It also assesses the extent to which teachers can identify appropriate strategies for integrating technology into the classroom to stimulate students' critical and self-critical thinking.

**Mediator variables.** *Perceived usefulness*. The perceived usefulness (PU) is the extent to which a person believes that using a particular system or instrument will improve his performance in his work field [40]. PU can also be defined as a person's perception of how using a specific technique will increase personal performance in solving work tasks [41]. Teachers' use of instructional strategies is valuable when it encourages interactions between all educational actors or allows them to model a real-life situation [22, 42]. The PU significantly affects the intention to use digital tools in education [43]. Moreover, usefulness mediates the relationship between ease of use and attitude [40, 44]. Perceived usefulness influences the adoption of modern technologies [45] due to strengthening the value of the final product. Perceived usefulness plays a decisive role in human behavior. This statement is based on the theory of reasoning that if a person perceives a particular activity as helpful in achieving specific results, that person adopts that activity. In addition, perceived usefulness has a more significant influence than other factors (i.e., enjoyment) when adopting new educational technologies [46].

In the case of the adoption of IBL/INT, the positive attitude of teachers can be determined through the PU, a belief arising from the competencies acquired in this field. In this study, the PU factor determines the extent to which teachers have tried implementing IBL/INT strategies, even if much effort was needed to prepare the lessons for teaching. It also evaluated the thoughts created by adopting each approach as a valuable and necessary method adopted by secondary teachers.

*Perceived enjoyment*. Perceived enjoyment (PE) is the degree to which an action involving technology is perceived as agreeable, separate from any anticipated consequences [47]. As an affective state, pleasure is an inherent part of the motivation (i.e., satisfaction) to be involved in a particular activity. In this respect, enjoyment can be seen as intrinsic motivation and affects the ease of using modern educational strategies. Teachers who consider a specific approach more enjoyable are more inclined to use it in practice than those who do not like it. Students recognized this attitude [48]. The pleasure also increases interest in that activity [49]. In brief, an activity the teacher is happy to conduct increases students' interest in the subject, motivation, and more efficient learning. Pleasure is linked to the positive emotions of the implication of the specific notions of nature scientists currently use in real life. However, please also depends on an activity's satisfaction and the stress level it produces [50]. A joyfully addressed activity increases involvement, and the proposed objectives are met [51]. STEM-related activities [52, 53] can generate satisfaction among students and teachers since these activities are perceived as complex [54]. As well, many forms of IBL (i.e., problem-based, project-based, and case-based teaching) [55] contributed to having more students with positive dispositions to choose to study various subjects [56].

In this study, the enjoyment factor assesses to what extent teachers are happy to use STEM teaching practices, whether they find the activities amusing or dull, whether they are attracted to their attention, and whether they find them interesting or enjoyable.

**Moderator variables.** *STEM teaching practices*: *IBL teaching practice*. Everything new is particularly challenging for students [57]. In such a context, IBL is perceived as an educational strategy [58] in which students follow methods and practices like those used by researchers to identify solutions to various problems [59]. Consequently, the IBL can be defined as discovering new causal relationships in which students devise hypotheses and test them by conducting experiments or using observations. The IBL solves problems and applies problem-solving skills [55, 60]. In addition, the IBL [37, 61] focuses on student's active participation and responsibility for discovering notions [14].

The educational objectives proposed will be meta-cognitive, affective, epistemic, or social [62] when teachers use the IBL approach. The rich teaching experience makes adopting IBL strategies [63] more effective, and teachers with more experience are more open to this new approach. Preparing teachers with less experience and providing practical guidance to implement IBL strategies in the classroom can positively change their attitude by reducing their reluctance. The advantages of this innovative approach, both in student understanding of content and developing their competencies, can be recognized and internalized. The confidence in performing an effective IBL activity can be increased [64]. In addition to the transfer of knowledge, teachers permanently influence students' attitudes toward science, either directly through the teaching process or involuntarily through the personal perspective they express. IBL strategies develop research experiences among students and help them think and act as researchers, positively influencing their attitude toward science and scientific research [65].

The teacher's attitude to the subject taught can affect the students' reaction and the effectiveness of the teaching and learning process. A teacher who feels comfortable with the subject taught and the approach adopted will devote more time to teaching, designing, and applying it creatively [18]. Previous studies showed that teachers' attitude was developed through training courses based on student-centered constructive methods [56–58], increasing their confidence in successfully implementing innovative strategies and increasing satisfaction in their work.

Even though the attitude of teachers is positive toward the IBL strategy, many consider it is challenging to implement it because the facilities for practices are inadequate. Moreover, teachers do not have enough teaching materials to use IBL [66, 67] in current activities. Preparing lessons take a long time, they cannot go through the school program entirely, and the

number of student classes is frequently excessively large [67]. Despite the difficulties, teachers recognize the advantages of IBL, such as increasing motivation for learning and understanding abstract concepts by their students.

### INT teaching practices

Implementing INT in classroom activities [68] can only be successful if teachers who teach INT subjects promote a positive attitude toward STEM. This attitude will increase the number of students interested in a career related to natural science and technologies [69, 70]. In this regard, an improvement in the attitude of teachers toward INT would lead to success in implementing this type of curriculum in classes.

For teachers, shifting from teaching a single subject to an interdisciplinary approach, such as the INT approach, is a significant challenge [52]. The interest in adopting INT teaching practice by secondary teachers must be improved. The primary solution following specialist literature is developing competencies in knowledge, skills, and attitude [71].

## Dependent variable

### Attitude

Attitude is a critical factor influencing the teacher's interest in a particular teaching style. Attitude can be defined as a positive or negative assessment of beliefs, intentions, or actions. Attitude is a psychological trend that assesses a particular entity as favorable or unfavorable [72]. The attitude of teachers influences what and how students learn [26], the primary factor affecting their acquisition and motivation for learning. In this framework, attitude can also be defined as a mental concept that describes favorable or unfavorable emotions toward a teaching strategy. A teacher with a positive attitude will also induce students to have a positive attitude toward studying a particular subject or behavior [50].

The attitude to a subject or behavior can be positive, negative, or neutral [73]. Some researchers stated that the attitude should be coupled with an interest [74, 75], and these two concepts are synonymous [76]. Other researchers view interest as a particular attitude that predicts future behavior [75, 77]. Therefore, interest is subordinate to attitude [78]. From another perspective, the two concepts are different from each other. General, impersonal, objective interpretations are decisive in determining attitudes to a subject, while subjective interpretations attached to the knowledge of the subject matter are essential for interest [79]. These two forms of personal approach are independent of each other. For example, a teacher can have a negative attitude toward a subject (i.e., discrimination) and at the same time to be interested in the subject in terms of its understanding, resorts, history, etc.

Teachers' attitudes towards adopting STEM teaching methods may or may not be a significant barrier to the integration of STEM education [13]. Teachers' attitudes towards adopting a teaching strategy are closely related to their knowledge of its use [80]. Firstly, teachers must know the didactic process very well to establish the stages of the lesson [80]. Secondly, unlike traditional approaches, in IBL and INT, the learner has a key role during the lesson [13]. For these reasons, the teacher needs to know the techniques to facilitate a learning process in which students have to take on various responsibilities (e.g., manager, researcher, and presenter roles). Also, STEM teachers need to have a comprehensive knowledge of the use of digital resources to run virtual and remote experiments. In conclusion, it can be stated that teachers' attitudes towards adopting STEM teaching practices depend mainly on their cognitive skills.

Studies on teachers' attitudes toward STEM teaching practices involving simultaneous affective and cognitive dimensions are rare [81]. The present study investigates the

relationships among the mediation moderation research model in response to this problem. The model comprised the competencies as a predictor, attitude as a dependent variable, usefulness and enjoyment as mediators, strategy style as a moderator variable and specialization as a control variable. The key factors related to each research model dimension were extracted based on literature evaluations to reach these objectives. A survey was built to examine relationships between variables. The questionnaire findings analyzed the theoretical research model using structural equation modeling. The mediation moderation model will help us find the algorithm to improve the exploration and understanding of real-life situations with respect to scientific rigor, stimulating the student's autonomy and creativity.

**The current study. Hypothesis summary.** The current study used a multiple mediation model to investigate the research relationships (see Fig 1). The present investigation demonstrates the existence of connections between the four latent factors of the research model.

The first working hypothesis starts from an obvious fact: teachers who know IBL or INT manifest an attitude or perspective on the strategy, positive or negative [82]. The relationship between competencies and attitudes is still unclear [83]. To have a more accurate picture of the factors influencing the teachers' attitudes, the association of competencies-attitude mediated by two affective factors, moderated by the didactic strategy used in the class and controlled by specialization, was investigated.

Attitudes and competencies define the usefulness and adoption of IBL and INT practices in the teaching activity, leading to a personalized experience. Teacher competencies are formed primarily through training. Attitudes toward STEM-training courses are directly related to teacher personality and internal and external motivational factors. Studies have shown that acquiring new skills through continuous teacher training can be the key factor in allowing them to develop their expertise by adopting technology-based teaching strategies [84] because of a positive attitude toward them. Therefore, it is natural to investigate whether the competencies directly influence teachers' attitudes.

The second hypothesis is that teachers have observed a relationship between competence, utility, and enjoyment of a particular teaching strategy. Otherwise, the perception of utility and enjoyment can influence attitudes to use certain teaching practices to improve learning outcomes [85]. IBL and INT practices can significantly influence teaching engagement [84].

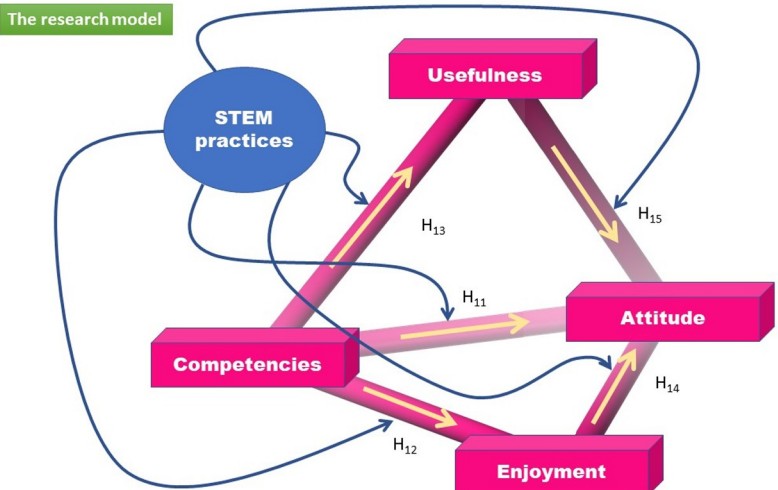

**Fig 1. Diagram of the research model.**

Teachers have a positive attitude toward strategies promoting interdisciplinary and investigative features [86], but they have little knowledge of each STEM domain [82].

The will, skill, tool (WST) model incorporates concepts that reflect attitude from the perspective of the tool user. Another helpful model for explaining the influence of perceived usefulness and enjoyment on the adoption of STEM teaching methods is Technology Acceptance Model (TAM) [23]. In TAM, teacher attitudes are influenced by the direct and indirect effects of perceived usefulness and perceived ease of use of educational technology. However, the TAM has proven to be inaccurate in explaining user behavior, especially under particular conditions of use [87]. This evidence challenges us to investigate whether PU can play a significant role in shaping teachers' attitudes towards IBL and INT [40, 44]. Another model for understanding behavior is the flow theory (FLT) [88]. Flow theory's applicability has been extended to online learning environments. Researchers have suggested that the success of STEM teaching depends on the teacher's attitude and ability to influence the learning process by using elements that generate enjoyment, such as making a product [89]. Indeed, as Rodríguez-Ardura and Meseguer-Artola (2017) stated, "continuance in e-learning is determined by flow through the mediating effects of positive emotions" [90]. The present study aims to synthesize WST, TAM, and FLT models in describing the relationships between secondary teachers' competencies and attitudes in the formal educational context under perceived usefulness and enjoyment.

Based on the research model, three main hypotheses were proposed to examine the predictors of the teachers' attitudes in two educational strategies settings:

*Hypothesis 1*. There are direct and positive relationships between Competencies and Attitude ($H_{11}$), Competencies and Enjoyment ($H_{12}$), Competencies and Usefulness ($H_{13}$), Enjoyment and Attitude ($H_{14}$) and Usefulness and Attitude ($H_{15}$), in the case of the total sample (a), IBL (b) and INT (c) teaching practices.

*Hypothesis 2*. The relationship between Competencies and Attitude is mediated by Enjoyment ($H_{21}$) and Usefulness ($H_{22}$) in the case of the total sample (a), IBL (b) and INT (c) teaching practices.

*Hypothesis 3*. The relationship between competencies and attitudes mediated by enjoyment and usefulness differs between IBL and INT teaching practices.

## Method

### Research design

The present study used a descriptive and cross-sectional research design with convenience and snowball sampling based on an online survey focused on research questions. This study was performed between 2020 and 2021. In summary, this study utilizes a partial least squares structural equation modelling (PLS-SEM) method to develop a model indicating the relationships among the independent variable (i.e., competencies), mediator variables (i.e., perceived usefulness and perceived enjoyment), and dependent variables (i.e., attitude). The demographic variables do not control the dependent variable, and the relationship between the independent and the dependent variables is only hypothetically causal. For this purpose, quantitative data were collected via an online questionnaire with demographic issues and multiple items for each research variable [23]. As a result, the researchers distributed the questionnaire to Romanian teachers who used IBL and INT on a cloud-based survey solution with real-time collaboration (i.e., Google Forms) to facilitate the completion and collection of data during the COVID-19 pandemic. The current study includes specialization into the research model as a

confounding variable. The standard stages for doing PLS-SEM were followed using SmartPLS software. Present findings can be used as a reference by educational institutions in formulating and implementing guidelines for STEM teaching practices.

## Sample

The target population of this study is Romanian teachers who used IBL and INT in their teaching process. The population of this research was composed of physics, biology, chemistry, technology, informatics, geography, and mathematics teachers. According to internal records of the Romanian Ministry of Education in the academic year 2019–20, this population was 29,676. Sample size estimation was calculated considering a confidence level = 95%, with an anticipated effect size = 0.01 and power of effect = 0.8 resulting [91] in a minimum sample size for model structure 288. The sample size of this study is 300 for each STEM teaching practice (Mage = 49, SD = 7.8, 87.7% women) and, thus, meets recommended guidelines.

## Participants

A total of 305 teachers from Romanian middle and high schools (5-12th grades) participated in this study. Therefore, six hundred ten individual responses from 305 STEM teachers were collected, then the data were examined to respect minimum requirements in quality and quantity [92]. Five samples did not meet the minimum standards in quality criteria [93] and were eliminated. Three hundred eligible answers have been collected for each key dimension of STEM teaching practice. Hence, the survey collected 600 eligible responses (S2 Table Raw database) through an Internet snowball sample design.

Teacher specialization can be divided into several categories (Table 1). An overwhelming majority (286/95%) have seen the IBL and INT presentation film since the beginning of the questionnaire (see S1 Table).

## Data collection

At the beginning of the present research, a methodological challenge occurred due to the necessity of recruiting 'hard-to-reach' populations for the study [93, 94]. Moreover, the Covid-19 Pandemic imposed various limitations [95] and changed the way of instruction [96], forcing all educational actors to adapt to online education [97] quickly. Consequently, collecting data from STEM teachers through traditional face-to-face survey administration was impossible.

Snowball sampling [98], as a non-probability sampling technique [99], is often used whenever there is no freely accessible list of the people under examination [100, 101]. Frequently snowball sampling is usually used in qualitative analyses [102]. However, quantitative studies also adopt the snowball sampling technique [103]. Virtual snowball sampling [93] is a recruitment method that involves social network participants [92] to contact specific target groups using interpersonal relations and connections between people [104]. This method collects online data from 'hard'-to-reach' populations [105] that cannot be directly observed [106]. A virtually recruited sample can improve the geographically diverse and speed of recruitment [107] and eliminate data missing [108, 109]. Before the online research tool development, the researchers studied all ethical issues (consent, privacy, and confidentiality) [110]. Data collecting started only after obtaining ethical approval, as in other studies [111–113].

A virtual snowball sampling method proper for behavioral sciences and linked with an online questionnaire as a matching tool [93] was selected to test the research hypothesis. Respondents voluntarily took part and submitted answers when they had completed the online questionnaire (see S1 Table. Survey questionnaire). This way, they can articulate more detailed answers to focused and accurate questions [93, 105].

**Table 1. Socio-demographic characteristics of the participants by specialization.**

| | Physics (N = 111) | Chemistry (N = 55) | Biology (N = 13) | Mathematics (N = 48) | Technology (N = 32) | Informatics (N = 14) | Geography (N = 27) | Total (N = 300) | *p* value |
|---|---|---|---|---|---|---|---|---|---|
| **Seniority** | | | | | | | | | 0.002 |
| Mean (SD) | 23.9 (10.2) | 26.2 (8.5) | 18.2 (7.6) | 23.7 (9.8) | 26.3 (5.7) | 19.5 (11.1) | 18.7 (10.0) | 23.6 (9.6) | |
| Range | 0.0–40.0 | 0.0–42.0 | 3.0–30.0 | 0.0–42.0 | 10.0–34.0 | 1.0–35.0 | 0.0–36.0 | 0.0–42.0 | |
| **Age** | | | | | | | | | < 0.001 |
| Mean (SD) | 49.5 (7.7) | 51.1 (6.3) | 45.2 (5.2) | 47.8 (9.3) | 51.6 (5.1) | 47.3 (10.5) | 44.2 (7.5) | 49.0 (7.8) | |
| Range | 23.0–64.0 | 26.0–64.0 | 34.0–53.0 | 21.0–69.0 | 36.0–61.0 | 26.0–59.0 | 22.0–55.0 | 21.0–69.0 | |
| **Degree** | | | | | | | | | 0.045 |
| Debutante | 12.0 (10.8%) | 2.0 (3.6%) | 0.0 (0.0%) | 3.0 (6.2%) | 0.0 (0.0%) | 3.0 (21.4%) | 4.0 (14.8%) | 24.0 (8.0%) | |
| Definitive | 4.0 (3.6%) | 2.0 (3.6%) | 2.0 (15.4%) | 2.0 (4.2%) | 0.0 (0.0%) | 2.0 (14.3%) | 0.0 (0.0%) | 12.0 (4.0%) | |
| Degree II | 13.0 (11.7%) | 3.0 (5.5%) | 3.0 (23.1%) | 5.0 (10.4%) | 2.0 (6.2%) | 1.0 (7.1%) | 4.0 (14.8%) | 31.0 (10.3%) | |
| Degree I | 82.0 (73.9%) | 48.0 (87.3%) | 8.0 (61.5%) | 38.0 (79.2%) | 30.0 (93.8%) | 8.0 (57.1%) | 19.0 (70.4%) | 233.0 (77.7%) | |
| **Residence** | | | | | | | | | 0.072 |
| Urban | 90.0 (81.1%) | 37.0 (67.3%) | 8.0 (61.5%) | 40.0 (83.3%) | 28.0 (87.5%) | 9.0 (64.3%) | 18.0 (66.7%) | 230.0 (76.7%) | |
| Rural | 21.0 (18.9%) | 18.0 (32.7%) | 5.0 (38.5%) | 8.0 (16.7%) | 4.0 (12.5%) | 5.0 (35.7%) | 9.0 (33.3%) | 70.0 (23.3%) | |
| **Training** | | | | | | | | | 0.285 |
| Continuous training | 37.0 (33.3%) | 18.0 (32.7%) | 1.0 (7.7%) | 16.0 (33.3%) | 4.0 (12.5%) | 3.0 (21.4%) | 10.0 (37.0%) | 89.0 (29.7%) | |
| Perfection courses | 44.0 (39.6%) | 19.0 (34.5%) | 9.0 (69.2%) | 21.0 (43.8%) | 17.0 (53.1%) | 8.0 (57.1%) | 12.0 (44.4%) | 130.0 (43.3%) | |
| Total training | 30.0 (27.0%) | 18.0 (32.7%) | 3.0 (23.1%) | 11.0 (22.9%) | 11.0 (34.4%) | 3.0 (21.4%) | 5.0 (18.5%) | 81.0 (27.0%) | |
| **Gender** | | | | | | | | | 0.083 |
| Female | 93.0 (83.8%) | 52.0 (94.5%) | 12.0 (92.3%) | 39.0 (81.2%) | 29.0 (90.6%) | 11.0 (78.6%) | 27.0 (100.0%) | 263.0 (87.7%) | |
| Male | 18.0 (16.2%) | 3.0 (5.5%) | 1.0 (7.7%) | 9.0 (18.8%) | 3.0 (9.4%) | 3.0 (21.4%) | 0.0 (0.0%) | 37.0 (12.3%) | |

A higher level of respondent focus on anonymous responses was ensured through self-administration [114] of the surveys. In the present study, the teachers who taught math, physics, biology, chemistry, technology, and informatics as part of the target group were contacted by emails or Facebook by authors. School inspectors for STEM disciplines were asked to introduce other teachers to participate. Some write an email invitation to the target group or support the Facebook author's invitation.

Data collection took sixty-six days (November 14, 2020, and January 18, 2021).

The survey distribution began by sending a message to the scholar inspectors for STEM disciplines. A URL for accessing the online questionnaire was included in this message, and it is still available (https://forms.gle/cSesCawKZm2vutR16). Using the snowball method, those inspectors recommended and helped recruit other research participants (e.g., teachers who taught math, physics, biology, chemistry, technology, and informatics) from their contacts. In this way, starting from the initial list, access to a more considerable number of STEM teachers has improved.

The context and purposes of the investigation were presented in the first part of the questionnaire. All ethical and personal data protection rules have been respected in collecting responses: no personal or other data have been collected, except for the type, age in education, degrees of teaching, and discipline taught. The authors also offered the participants a short, simple, and relevant explanation of the purpose and method of the study to start the survey. Additionally, the authors used a 29-minute video presentation describing a practical example of implementing STEM learning at the beginning of the questionnaire. The video presentation summarized IBL and INT usage cases in different fields of study. The questions were

standardized, with pre-coded and focused answer choices. Thus, the questionnaire, with the presentation video, was transmitted through the communication channels of the teachers. The teachers' Facebook groups were also used. The questionnaire had a part with open questions, which allowed the authors to summarize the advantages enjoyed by IBL and INT.

## Questionnaire development

At present, students are no longer oriented toward studying exact sciences, although the society we live in is increasingly using technology. According to the Organization for Economic Cooperation and Development (OECD), there are three times more graduates of human, social or legal disciplines than science, technology, engineering, and mathematics [115]. This trend has been unchanged for more than ten years [115]. In addition, more students enrolled at a high school with a technical profile abandon the school than those enrolled at a high school with a humanist profile [116]. Although more people are increasingly dependent on technology, and the labor market demands an increasing number of specialists in technical fields, an ever-smaller number of students are interested in studying the science of nature or technology. The vocational orientation of the students starts in school, and teachers play a crucial role [117]. From these considerations, this study investigated the attitude of teachers representing the human resource that provides the educational process. In addition, IBL and INT can be used successfully in teaching the science of nature [13, 61, 118]. Moreover, if the teacher participates as an education facilitator, the learners' results significantly improve [119].

The main purpose of this study was to examine teachers' level of knowledge about INT and IBL teaching practices and their attitude toward applying these methods in current teaching. The questionnaire was developed in a seven-point Likert scale format to answer based on the study framework and hypotheses as precisely as possible. The questionnaire was structured in three sections: the first section collects specific information about respondents to determine the distribution of answers according to specialization, seniority, age, degree category, residence, training type, and gender. The second section covers the specific items of the latent factors of the research model competence, perception of usefulness, the perception of enjoyment and attitude. Experts in Natural Sciences and Educational Sciences contributed to the questionnaire design. A professional translation from Romanian into English was performed. The questionnaire application was approved by the Bucharest University Ethics Committee and complied with all rules on protecting the personal data of the research participants. No personal data were collected, and the answers received were used only for research purposes.

## Measurements

**Competences.** The competencies regarding knowledge of the pedagogical content of IBL and the INT teaching practices were assessed with the items adapted to the scale proposed by Chuang, Weng, & Huang [120]. The initial scale contains eight items for each topic addressed (IBL/INT). Items measured two dimensions: content (i.e., "*I can identify topics appropriate for teaching through IBL/INT to help students better understand the problematic content*") and personality (i.e., "*I can use IBL/INT to stimulate learners to observe, explore and investigate*") (see S1 Table). All items were rated on a 7-point Likert scale ranging from 1 (Total disagreement) to 7 (Total agreement). Four items do not meet quality requirements of Exploratory Factor Analyses (EFA). Consequently, these items were removed from the scale and rebuilt evaluation scale. A Cronbach alpha coefficient was calculated for the Competencies scale, consisting of SI12, SI13, SI14, and SI16 (Table 2).

**Usefulness.** Items related to the usefulness dimension were self-developed. This factor contains four items for each topic (IBL/INT). Items measured two dimensions: importance

**Table 2. Factor loadings from exploratory factor analysis for total sample (N = 600).**

| Factor Loadings | | | | | |
|---|---|---|---|---|---|
| | Factor | | | | |
| | Competences | Usefulness | Attitude | Enjoyment | Communality |
| SI12 | 0.90 | | | | 0.74 |
| SI13 | 0.72 | | | | 0.68 |
| SI16 | 0.72 | | | | 0.67 |
| SI14 | 0.68 | | | | 0.56 |
| SI71 | | 0.87 | | | 0.84 |
| SI63 | | 0.77 | | | 0.50 |
| SI72 | | 0.69 | | | 0.71 |
| SI64 | | 0.63 | | | 0.45 |
| SI35 | | | 0.85 | | 0.76 |
| SI36 | | | 0.78 | | 0.59 |
| SI34 | | | 0.73 | | 0.67 |
| SI33 | | | 0.59 | | 0.50 |
| SI53r | | | | 0.92 | 0.76 |
| SI54r | | | | 0.76 | 0.75 |
| SI55 | | | | 0.48 | 0.71 |

*Note.* 'Principal axis factoring' extraction method was used in combination with an 'oblimin' rotation

(i.e., "*Because it seems important, I would put a lot of effort to promote IBL/INT activities*") and valuable (i.e., "*I would be inclined to apply IBL/INT because I find it a valuable strategy for all educational actors*") (see S1 Table). All items were rated on a 7-point Likert scale ranging from 1 (Total disagreement) to 7 (Total agreement). A Cronbach alpha coefficient was calculated for the Usefulness scale, consisting of SI71, SI63, SI72, SI64 (Table 2).

**Enjoyment.** Four self-developed items measured teachers' enjoyment relative to the teaching act with STEM and IBL strategies. Items measured one dimension: enjoy (i.e., "*IBL/INT activity is very interesting*") (see S1 Table). All items were rated on a 7-point Likert scale ranging from 1 (Total disagreement) to 7 (Total agreement). One item does not meet quality requirements of Exploratory Factor Analyses (EFA). Consequently, this item was removed from the scale and rebuilt evaluation scale. A Cronbach alpha coefficient was calculated for the Enjoyment scale, consisting of SI53r, SI54r, and SI56 (Table 2).

**Attitude.** Ten items measured teachers' attitude on using the IBL practice [121] and INT [9]. Items were measuring one dimension regarding cognitive attitude (i.e., "*IBL/INT strategy is diverse and has various paths*" and "IBL/INT *leads to the stimulation of learners' autonomy*" (see S1 Table). All items were rated on a 7-point Likert scale ranging from 1 (Total disagreement) to 7 (Total agreement). Six items do not meet quality requirements of Exploratory Factor Analyses (EFA). Consequently, these items were removed from the scale and rebuilt evaluation scale. A Cronbach alpha coefficient was calculated for the Attitude scale, consisting of SI33, SI34, SI35, SI36 (Table 2).

## Data analysis

Various methods were conducted to determine the reliability and validity of the survey. First, questionnaire face validity was determined by educational experts' contribution to the item's development and by pretesting among Bucharest University teachers. Then, an item analysis was made to analyze the relevance of each item [122]. All values of the item-total correlations

were higher than the cut-off criterion of 0.20 [123]. All items that meet the quality requirements were retained in the questionnaire.

To assess the study model in each STEM practice using PLS-SEM and to match the findings of the estimated path coefficients, this study followed a three-stage approach to estimate the measurement models, structural models, and multigroup analysis [124]. Exploratory factor analysis (EFA) was applied to total sample corresponding to IBL and INT teaching strategies (N = 600) to provide the construct validity of the questionnaire [120, 125]. Exploratory factor analysis (EFA) [34] was conducted for 26 variables (see S1 Table) using parallel analysis for determining the number of factors to retain with the principal axis factoring method and Oblim rotation. EFA were performed to examine the relationship between the questionnaire items and the proposed theoretical concepts. This determines factorial validity, which is a part of construct validity [126]. The factor loadings were interpreted by taking the absolute value of each loading and implementing the criterion suggested by Comrey and Lee [127]. Values greater than 0.71 are considered excellent, values between 0.63 and 0.71 are excellent, values between 0.55 and 0.63 are good, values between 0.45 and 0.55 are fair, and values between 0.32 and 0.45 are poor. Tabachnick and Fidell [128] also recommend that 0.32 be the minimum threshold to identify significant factor loadings. Kaiser-Meyer-Olkin (KMO) (>.60 is adequate) and Bartlett tests (significant at the 0.05 level) were analyzed to know whether the data set is suitable for the factor analysis. The results of these tests met the conditions (KMO = 0.95) and Bartlett tests were significant at the 0.05 level, therefore the EFA procedures were started. EFA supported the pre-defined structure of the four constructs (competencies, enjoyment, usefulness, and attitude). All rotated factor loadings associated with corresponding dimensions and all commonalities were greater than .40 (see Table 2).

Briefly, the sample size for exploratory factor analysis was 600, and the initial number of variables included was 26. Therefore, the sample size is adequate to produce reliable outcomes [129]. Factor 1 (Attitude) accounted for IBL teaching practices: 18.3% of the variance with an eigenvalue of 8.37. Factor 2 (Competencies) accounted for 18.2% of the variance. Factor 3 (Usefulness) accounted for 18.1% of the variance. Factor 4 (Enjoyment) accounted for 14% of the variance. The four-factor model accounted for 68.7% of the total variance in the data. The factor analysis loadings and the communality estimate, or each item are presented in Table 2. There were no variables with a low communality (< 0.20) or cross-loadings, which indicates that the factor structure is uncomplicated and easy to explain. Moreover, each factor had at least three significant loadings (> 0.40), which is revealing of a solid and consistent factor [129].

Eleven items were excluded from the EFA analyses (the cut-off limit > 0.40) [125], and further analysis was performed with the remaining 15 items.

Then, a confirmatory factor analysis (CFA) using SmartPLS3 with two separate sample corresponding to IBL and INT teaching practice was performed. Ratio between Chi-square ($\chi2$) and degrees of freedom (*Df*), comparative fit index (CFI), Tucker-Lewis fit index (TLI), Root mean square error of approximation (RMSEA), standardized root mean square residual (SRMR) were quality indexes taking into consideration. CFA results indicated an adequate model fit for IBL/INT: $\chi2$/*Df* = 2.26/2.88, CFI = 0.96/0.95, TLI = 0.95/0.94, RMSEA = 0.06/0.04, SRMR = 0.04/0.04. Then, the Cronbach alpha coefficients were calculated for each factor of each teaching practice to assess the reliability (internal consistency) of the survey instrument.

In the last part, a partial least squares structural equation modeling (PLS-SEM) model [130, 131] using SmartPLS version 4 software was conducted to determine whether the latent variables (Attitude, Usefulness, Competencies, and Enjoyment) adequately describe the data. The following fit index was used to assess the model fit: standardized root mean square residual (SRMR) [132, 133]. The effect size of the predictor dimensions was assessed using $f^2$ effect

size. According to Cohen (1988), $f^2$ a value of 0.02, 0.15, and 0.35 represent small, medium, and large effects [134].

Finally, invariance tests were conducted to test whether the measurement model met the metric invariance criteria across strategy styles. Measurement invariance composite (MICOM) results (S3 Table) based on three steps (configural invariance, compositional invariance, & equality of composite mean values and variances) [135] revealed that partial measurement invariance had been established. Moreover, the permutation [136] outputs infirmed that the possible differences are a consequence of the STEM teaching style. Exploratory factor analysis (EFA), Confirmatory Factor Analyses (CFA) and Partial least squares structural equation modeling (PLS-SEM) analyses were performed using the Jamovi, Intelectus statistics [137], and SmartPLS version 4 software packages.

## Results

### Data normality

Evaluation of the skewness and kurtosis distribution were used to check the normality of data. The study constructs did not exhibit a normal distribution, as in some situation's skewness varied over values from -2 to +2, and kurtosis values ranged over values from -3 to +3 (see Table 3). PLS-SEM algorithms are generally used in the case of non-normal data [138]. Consequently, even if some of the item responses do not represent a normal distribution, further analysis was performed using PLS-SEM.

### Common method bias

Questionnaire data based on same-respondent responses may lead to common method bias. Any variable with an $R^2 > 0.90$ can contribute to multicollinearity in the SEM model [122]. Consequently, the variance inflation factor (VIF) [139] was verified if multicollinearity is or not severe. All inner VIF values are smaller than the cut-off value of 3.3 [135], so the multicollinearity is not severe.

### Descriptive statistics and correlations

Summary statistics were calculated for the Competencies, Usefulness, Enjoyment, and Attitude scales. There was little difference in the mean scores of all the items, as the values fluctuated from 5.39 to 6.15. The results of descriptive statistics and corrected item-total correlation the research model dimensions are presented in Table 3. All corrected item-total correlations are scored positively and more than 0.3, which was in line with recommendations [140]. The correlation coefficients of this study are reported in Table 4. As can be observed in Table 4, significant and positive correlations were found between each of the latent variables for each case study. Briefly, teacher competencies and attitudes were related positively to usefulness, and enjoyment, as predicted by the research model (see Table 4).

### Reliability and validity of the measurement models

The results of the factor loading (standardized regression weights) indicate that the magnitude for all items is above the satisfactory benchmark of 0.71 (Table 5).

Also, the data are considered to be consistent because the value of the Cronbach's alpha, rho_A coefficient and composite reliability (CR) (Table 6) lies within the acceptable limits of above 0.7 [141]. The Average Variance Extracted (AVE) for the research dimensions corresponding to both samples IBL/INT was greater than the threshold 0.50 (see Tables 5 and 6), so the convergent validity of constructs was fulfilled [142]. The off-diagonal values shown in

**Table 3. Descriptive statistics and corrected item-total correlations of the 14 items corresponding to Competencies, Usefulness, Enjoyment, and Attitude scales (N1 = 300 (IBL) and N2 = 300 (INT)).**

| Factor/Items | Strategy | Mean | SD | Corrected Item-Total Correlation | Skewness | | Kurtosis | |
|---|---|---|---|---|---|---|---|---|
| | | | | | Skewness | SE | Kurtosis | SE |
| **Competencies** | | | | | | | | |
| SI12 | IBL | 5.93 | 0.90 | 0.79 | -1.75 | 0.14 | 6.27 | 0.28 |
| | INT | 5.83 | 0.87 | 0.76 | -1.47 | 0.14 | 4.54 | 0.28 |
| SI13 | IBL | 6.03 | 0.78 | 0.73 | -1.62 | 0.14 | 6.89 | 0.28 |
| | INT | 5.88 | 0.80 | 0.79 | -1.46 | 0.14 | 4.16 | 0.28 |
| SI14 | IBL | 5.75 | 0.98 | 0.67 | -1.53 | 0.14 | 4.18 | 0.28 |
| | INT | 5.72 | 0.97 | 0.71 | -1.22 | 0.14 | 2.17 | 0.28 |
| SI16 | IBL | 6.15 | 0.88 | 0.73 | -2.30 | 0.14 | 9.51 | 0.28 |
| | INT | 6.00 | 0.80 | 0.77 | -1.41 | 0.14 | 3.83 | 0.28 |
| **Usefulness** | | | | | | | | |
| SI71 | IBL | 5.86 | 0.94 | 0.78 | -1.62 | 0.14 | 4.60 | 0.28 |
| | INT | 5.73 | 0.99 | 0.82 | -1.41 | 0.14 | 2.92 | 0.28 |
| SI63 | IBL | 5.39 | 1.22 | 0.60 | -1.09 | 0.14 | 1.15 | 0.28 |
| | INT | 5.43 | 1.11 | 0.73 | -1.04 | 0.14 | 1.44 | 0.28 |
| SI64 | IBL | 5.72 | 0.98 | 0.65 | -1.23 | 0.14 | 1.80 | 0.28 |
| | INT | 5.69 | 0.99 | 0.60 | -1.37 | 0.14 | 2.71 | 0.28 |
| SI72 | IBL | 5.84 | 0.93 | 0.73 | -1.82 | 0.14 | 5.97 | 0.28 |
| | INT | 5.70 | 1.07 | 0.75 | -1.50 | 0.14 | 3.19 | 0.28 |
| **Enjoyment** | | | | | | | | |
| SI53r | IBL | 5.71 | 1.23 | 0.73 | -1.47 | 0.14 | 2.39 | 0.28 |
| | INT | 5.62 | 1.19 | 0.81 | -1.28 | 0.14 | 1.72 | 0.28 |
| SI54r | IBL | 5.96 | 1.07 | 0.79 | -1.79 | 0.14 | 4.42 | 0.28 |
| | INT | 5.80 | 1.09 | 0.79 | -1.54 | 0.14 | 3.02 | 0.28 |
| SI55 | IBL | 5.87 | 0.96 | 0.74 | -1.59 | 0.14 | 4.22 | 0.28 |
| | INT | 5.78 | 1.00 | 0.73 | -1.63 | 0.14 | 4.05 | 0.28 |
| **Attitude** | | | | | | | | |
| SI33 | IBL | 5.85 | 0.97 | 0.64 | -1.88 | 0.14 | 5.39 | 0.28 |
| | INT | 5.96 | 0.82 | 0.67 | -1.34 | 0.14 | 3.23 | 0.28 |
| SI34 | IBL | 5.80 | 0.91 | 0.71 | -1.10 | 0.14 | 2.02 | 0.28 |
| | INT | 5.72 | 0.95 | 0.75 | -1.50 | 0.14 | 3.72 | 0.28 |
| SI35 | IBL | 5.85 | 0.88 | 0.75 | -1.34 | 0.14 | 2.99 | 0.28 |
| | INT | 5.79 | 0.87 | 0.82 | -1.45 | 0.14 | 3.81 | 0.28 |
| SI36 | IBL | 5.66 | 1.00 | 0.66 | -1.28 | 0.14 | 2.32 | 0.28 |
| | INT | 5.69 | 0.95 | 0.76 | -1.33 | 0.14 | 2.51 | 0.28 |

Table 7 indicate the correlations between the factors. The diagonal values report the square values of AVEs, proving AVE value exceed the square correlations between the latent variables [143]. The confirmation of discriminant validity of the analysis is also validated because the Heterotrait-Monotrait Ratio of Correlations (HTMT) values were lower or equal than the limits of 0.90 (Table 7) [144].

## Structural model: Hypotheses tests

A structural model [145] with the consequences postulated in our PLS-SEM model was investigated. Of the demographic variables involved as controls (i.e., age, gender, seniority, residence, degree, training, specialization), our PLS-SEM model outcomes corresponding to the

**Table 4. Correlations between scores of the individual-level predictors (*competencies, usefulness, and enjoyment*) and effect (*attitude*).** Case study: IBL ($N_1$ = 300)/INT ($N_2$ = 300).

|  | 1 | 2 | 3 | 4 |
|---|---|---|---|---|
| 1. Competencies | — | 0.63[a]***/0.74[b]*** | 0.59 [a]***/0.63 [b]*** | 0.72 [a]***/0.69 [b]*** |
| 2. Usefulness |  | — | 0.69 [a]***/0.62 [b]*** | 0.61 [a]***/0.65 [b]*** |
| 3. Enjoyment |  |  | — | 0.58 [a]***/0.59 [b]*** |
| 4. Attitude |  |  |  | — |

*Note*: [a] = IBL;

[b] = INT;

*** $p < 0.001$

**Table 5. CFA factor loadings, t statistics and significance level for IBL and INT teaching strategies.**

| IBL (b) | | | |
|---|---|---|---|
|  | Loadings | T statistics (\|O/STDEV\|) | *p* values |
| SI12 <- Competencies | 0.89 | 39.65 | 0.000 |
| SI13 <- Competencies | 0.85 | 18.63 | 0.000 |
| SI14 <- Competencies | 0.81 | 23.21 | 0.000 |
| SI16 <- Competencies | 0.86 | 30.98 | 0.000 |
| SI33 <- Attitude | 0.79 | 20.23 | 0.000 |
| SI34 <- Attitude | 0.86 | 44.61 | 0.000 |
| SI35 <- Attitude | 0.88 | 41.55 | 0.000 |
| SI36 <- Attitude | 0.79 | 22.81 | 0.000 |
| SI53r <- Enjoyment | 0.85 | 19.79 | 0.000 |
| SI54r <- Enjoyment | 0.91 | 47.08 | 0.000 |
| SI55 <- Enjoyment | 0.91 | 73.22 | 0.000 |
| SI63 <- Usefulness | 0.74 | 13.31 | 0.000 |
| SI64 <- Usefulness | 0.79 | 22.99 | 0.000 |
| SI71 <- Usefulness | 0.91 | 62.12 | 0.000 |
| SI72 <- Usefulness | 0.89 | 44.22 | 0.000 |
| INT (c) | | | |
|  | Original sample (O) | T statistics (\|O/STDEV\|) | *p* values |
| SI12 <- Competencies | 0.88 | 35.99 | 0.000 |
| SI13 <- Competencies | 0.89 | 42.76 | 0.000 |
| SI14 <- Competencies | 0.82 | 25.62 | 0.000 |
| SI16 <- Competencies | 0.88 | 42.53 | 0.000 |
| SI33 <- Attitude | 0.81 | 22.88 | 0.000 |
| SI34 <- Attitude | 0.87 | 38.50 | 0.000 |
| SI35 <- Attitude | 0.91 | 55.66 | 0.000 |
| SI36 <- Attitude | 0.87 | 40.20 | 0.000 |
| SI53r <- Enjoyment | 0.91 | 56.11 | 0.000 |
| SI54r <- Enjoyment | 0.91 | 43.61 | 0.000 |
| SI55 <- Enjoyment | 0.89 | 44.98 | 0.000 |
| SI63 <- Usefulness | 0.84 | 24.67 | 0.000 |
| SI64 <- Usefulness | 0.75 | 12.60 | 0.000 |
| SI71 <- Usefulness | 0.92 | 86.93 | 0.000 |
| SI72 <- Usefulness | 0.88 | 26.79 | 0.000 |

**Table 6. Reliability and validity for IBL and INT teaching strategies.**

| Factor | Cronbach's alpha | | rho_A | | Composite Reliability CR. | | Average Variance Extracted AVE | |
|---|---|---|---|---|---|---|---|---|
| | IBL | INT | IBL | INT | IBL | INT | IBL | INT |
| Competencies | 0.87 | 0.89 | 0.87 | 0.90 | 0.91 | 0.92 | 0.73 | 0.75 |
| Usefulness | 0.85 | 0.87 | 0.88 | 0.89 | 0.90 | 0.91 | 0.70 | 0.72 |
| Enjoyment | 0.87 | 0.89 | 0.90 | 0.89 | 0.92 | 0.93 | 0.79 | 0.81 |
| Attitude | 0.85 | 0.89 | 0.86 | 0.89 | 0.90 | 0.92 | 0.69 | 0.75 |

total sample showed that none of these variables had a significant impact on attitude. Referring to the model quality for the total sample/IBL/INT, the study results showed an $R^2$ value of 0.58/0.59/0.59 for attitude, 0.42/0.45/0.39 for enjoyment and 0.48/0.42/0.54 for usefulness, higher than the 0.10 threshold [146]. The Stone-Geisser blindfolding sample reuse analysis shows relevance $Q^2$ values greater than 0 [124] for total sample and for both IBL/INT samples. Therefore, attitude ($Q^2$ = 0.52 (total sample)/0.53 (IBL)/0.52 (INT)), enjoyment ($Q^2$ = 0.41 (total sample)/0.44 (IBL)/0.38 (INT)) and usefulness ($Q^2$ = 0.47 (total sample)/0.41 (IBL)/0.53 (INT)) are successfully predicted by the research model. Moreover, the standardized root mean square residual (SRMR) of the total sample (0.06), IBL (0.07) and INT models (0.06), which is below the 0.08 cut-off [124, 146], indicated a good model fit for all case studies. The predicted paths are all significant, as displayed in Fig 2A and 2B.

The standardized coefficients present in Fig 2 emphasize that IBL/INT attitude is predicted by usefulness (β = 0.14/0.14), competencies (β = 0.49/0.46), and enjoyment (β = 0.22/0.26).

**Direct regression results. Correspondence with H1.** Direct regressions were examined based on an alpha value of 0.05. The brief outcomes of direct regression analyses and the effects size evaluation of the exogenous latent variables on the endogenous variable in their direct relationships are presented in Table 8 for total sample (a), IBL (b) and INT (c).

The Competencies (COM) significantly predicted Attitude (ATT) (see Table 8). This finding indicates that a one-unit increase in COM will increase the expected value of ATT by 0.47/0.49/0.47 units in the case of total sample/IBL/INT. Competencies (COM) significantly predicted Enjoyment (ENJ). Therefore, a one-unit increase in COM will increase the expected value of ENJ by 0.65/0.67/0.62 units. Competencies (COM) significantly predicted Usefulness (USE), indicating a one-unit increase in COM will increase the expected value of USE by 0.69/0.65/0.73 units in the case of total sample/IBL/INT. Enjoyment (ENJ) significantly predicted Attitude (ATT). Therefore, a one-unit increase in ENJ will increase the expected value of ATT by 0.24/0.22/0/26 units in the case of total sample/IBL/INT. Usefulness (USE) significantly predicted ATT, indicating a one-unit increase in USE will increase the expected value of ATT by 0.14 units for all case studies. As can be observed, the findings regarding direct regressions for IBL and INT are similar to the whole sample (see Table 8). These results confirm that hypothesis $H_1$ was supported.

**Table 7. Discriminant validity for IBL ([a]) and INT ([b]) teaching strategies.**

| | 1. | 2. | 3. | 4. |
|---|---|---|---|---|
| 1. Attitude | 0.83[a]/0.86[b] | | | |
| 2. Competencies | 0.73[a]/0.72[b] | 0.85[a]/0.87[b] | | |
| 3. Enjoyment | 0.65[a]/0.64[b] | 0.67[a]/0.62[b] | 0.89[a]/0.89[b] | |
| 4. Usefulness | 0.62[a]/0.65[b] | 0.65[a]/0.73[b] | 0.73[a]/0.69[b] | 0.83[a]/0.85[b] |

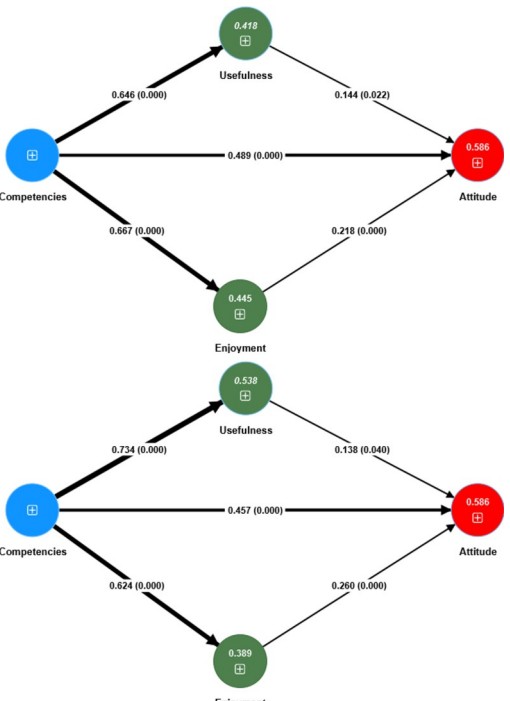

**Fig 2.** Attitude Structural Equation Model for IBL (a) and INT (b).

**Mediation model results. Correspondence with H2.** The suggested mediational models were tested to validate the second hypothesis. A more complex model consisting of indirect effects between competencies and attitude was looking to demonstrate (see Table 9). Briefly, the mediation test was conducted to determine whether USE or ENJ mediated the relationship between COM and ATT. The brief outcomes of mediation examined based on an alpha of 0.05 are presented in Table 9 (a (for average), b (IBL) and c (INT)). The direct effect between COM and ATT was significant (Table 8), suggesting that complete mediation by USE or ENJ did not occur, although some partial mediation was still present.

Partial mediation was examined using the indirect and total effects of USE or ENJ on the relationship between COM and ATT for each teaching practice (Table 9B and 9C) and for total sample (Table 9A). The indirect effect of USE on the relationship of ATT regressed on COM was significant for both IBL and INT. Therefore, a one-unit increase in COM, based on its effect on USE, will increase the expected value of ATT by 0.09 units for IBL and 0.10 units for INT. The indirect effects of USE on the relationship of ATT regressed on COM were low but statistically significant. A one-unit increase in COM, based on its effect on ENJ, will increase the expected value of ATT by 0.15 units for IBL and 0.16 units for INT. The total effect of COM on ATT was significant, indicating a one-unit increase in COM will increase the expected value of ATT by 0.73 units for IBL and by 0.72 for INT. Since the indirect and total effects were significant for IBL and INT, partial mediation was supported for both teaching practices (Table 9B and 9C). Consequently, hypothesis $H_2$ was supported.

**Multigroup model results. Correspondence with H3.** Additionally, the multigroup (MGA) comparison test results show non-significant differences between the IBL and INT as distinct groups for the relationship between COM and ATT, COM and ENJ, COM and USE, ENJ and ATT and USE and ATT (Table 10). Therefore, hypothesis H3 was not supported.

**Table 8.** Specific direct effects and the effects size evaluation of the exogenous latent variables on the endogenous variable in their direct relationships for the total sample (a), IBL (b) and INT (c) teaching practices.

| Hypothesis | Path | Path coefficients | Standard deviation (STDEV) | T statistics (\|O/ STDEV\|) | p values | $f^2$ effect size | Effect? | Hypothesis results |
|---|---|---|---|---|---|---|---|---|
| | Total sample (a) | | | | | | | |
| H11a | Competencies -> Attitude | 0.47 | 0.05 | 9.97 | 0 | 0.25 | Medium | Supported |
| H12a | Competencies -> Enjoyment | 0.65 | 0.04 | 17.87 | 0 | 0.71 | Large | Supported |
| H13a | Competencies -> Usefulness | 0.69 | 0.03 | 21.55 | 0 | 0.91 | Large | Supported |
| H14a | Enjoyment -> Attitude | 0.24 | 0.05 | 4.95 | 0 | 0.06 | Small | Supported |
| H15a | Usefulness -> Attitude | 0.14 | 0.05 | 2.62 | 0.004 | 0.02 | Small | Supported |
| | IBL (b) | | | | | | | |
| H11b | Competencies -> Attitude | 0.49 | 0.07 | 7.46 | 0 | 0.29 | Medium | Supported |
| H12b | Competencies -> Enjoyment | 0.67 | 0.05 | 13.92 | 0 | 0.80 | Large | Supported |
| H13b | Competencies -> Usefulness | 0.65 | 0.05 | 12.06 | 0 | 0.72 | Large | Supported |
| H14b | Enjoyment -> Attitude | 0.22 | 0.07 | 3.36 | 0 | 0.05 | Small | Supported |
| H15b | Usefulness -> Attitude | 0.14 | 0.07 | 2.01 | 0.022 | 0.02 | Small | Supported |
| | INT (c) | | | | | | | |
| H11c | Competencies -> Attitude | 0.46 | 0.07 | 6.46 | 0 | 0.22 | Medium | Supported |
| H12c | Competencies -> Enjoyment | 0.62 | 0.05 | 12.00 | 0 | 0.64 | Large | Supported |
| H13c | Competencies -> Usefulness | 0.73 | 0.04 | 19.76 | 0 | 1.17 | Large | Supported |
| H14c | Enjoyment -> Attitude | 0.26 | 0.07 | 3.84 | 0 | 0.08 | Small | Supported |
| H15c | Usefulness -> Attitude | 0.14 | 0.08 | 1.76 | 0.04 | 0.02 | Small | Supported |

The multiple mediation model underlined that competencies exert a direct effect on attitude and indirect effects through usefulness and enjoyment. As expected, the high level of competencies enhances the perceived usefulness and enjoyment of using a learning strategy, contributing to a higher attitude to adopt it during the educational process.

**Table 9.** Specific indirect effects for the total sample (a), IBL (b) and INT (c) teaching practices.

| Hypothesis | Path | Indirect effect coefficient | Standard deviation (STDEV) | T statistics (\|O/ STDEV\|) | p values | Hypothesis results |
|---|---|---|---|---|---|---|
| | Total sample (a) | | | | | |
| H21a | Competencies -> Enjoyment -> Attitude | 0.15 | 0.03 | 4.65 | 0.001 | Partial mediation |
| H22a | Competencies -> Usefulness -> Attitude | 0.10 | 0.04 | 2.58 | 0.005 | Partial mediation |
| | IBL (b) | | | | | |
| H21b | Competencies -> Enjoyment -> Attitude | 0.15 | 0.04 | 3.28 | 0.001 | Partial mediation |
| H22b | Competencies -> Usefulness -> Attitude | 0.09 | 0.05 | 1.96 | 0.025 | Partial mediation |
| | INT (c) | | | | | |
| H21c | Competencies -> Enjoyment -> Attitude | 0.16 | 0.05 | 3.53 | 0.001 | Partial mediation |
| H22c | Competencies -> Usefulness -> Attitude | 0.10 | 0.06 | 1.75 | 0.040 | Partial mediation |

**Table 10. PLS-MGA results.**

| Path | Path Coefficients-diff (IBL—INT) | *p*-Value original 1-tailed (IBL vs INT) | *p*-Value new (IBL vs INT) | Difference established? |
|---|---|---|---|---|
| Competencies -> Attitude | 0.03 | 0.369 | 0.369 | No |
| Competencies -> Enjoyment | 0.04 | 0.271 | 0.271 | No |
| Competencies -> Usefulness | -0.09 | 0.913 | 0.087 | No |
| Enjoyment -> Attitude | -0.04 | 0.67 | 0.330 | No |
| Usefulness -> Attitude | 0.01 | 0.479 | 0.479 | No |

## Discussions

### Practical implications

It is essential to explore the differential impact of STEM on the relationship between teachers' competencies and their attitude regarding integrated STEM. It offers valuable data to teachers who need to change educational acts according to their teaching practice's specific features [71]. Kelley and Knowles (2016) indicated that STEM educators lack a cohesive understanding of STEM concepts [147]. Integrated STEM requires transdisciplinary knowledge and skills [148, 149]. An integrated STEM curriculum [150, 151] could help solve the challenges of modern education. Also, the researchers have shown that perceived usefulness [152] and perceived enjoyment [153] influence the motivation to use STEM education [154]. Anything that teachers find real, useful and enjoyable through their engagement in specific activities is a stimulus for developing their competencies [155] and induces a positive attitude towards the educational process. Consequently, the interest in understanding the mediating roles of utility and enjoyment in the relationship between teachers' competencies and attitudes towards each STEM practice is justified to be investigated to forecast if this strategy will be successful.

The present research aimed to explore the direct predictors of the teachers' attitude to use STEM practices in educational activities, testing hypothesis H1 and indirect predictors testing H2. Also, we compared two STEM methods testing hypothesis H3.

### Direct relationships between antecedents and teacher attitude

Teacher attitudes were directly related to teachers' competencies, enjoyment and usefulness of both facets of STEM practices. These outputs complement other studies [4, 139, 140] and contribute to STEM curriculum development. The attitude influenced by the teachers' competencies decisively participates in successfully adopting a teaching strategy [156–158]. These results can be explained by defining the role of each teaching practice. IBL is a strategy related to direct interaction with the class and the teaching act itself. INT is a much broader concept and a plan suited to multidisciplinary subjects. These findings can be justified by considering the intrinsic motivational role of daily observations in the perceived usefulness and enjoyment of IBL and INT.

### Indirect relationships between competencies and teacher attitude

Education researchers have extensively studied the relationship between attitude and teachers' competencies. In particular, the relationship between attitude and competencies mediated by different aspects correlated with technology was studied to increase the probability of teachers embracing change [159]. However, the moderator factors that may intensify or diminish the relationship between the two constructs have not been fully identified. Some researchers have proven that ease of use influences the relationship between competence and behavioral intent to use correlated with attitude. Other researchers have found that the two dimensions are

mediated by the sense of efficacy [160] if we refer to the attitude towards inclusion or to the level of online teaching intentions if we refer to the attitude towards online teaching [161]. The present study highlights the simultaneous mediating influence of enjoyment and usefulness of a new teaching strategy that shares commonalities with adopting digital technologies. However, the results of the research show that perceived Usefulness (USE) plays a mediating role between competencies (COM) and attitudes (ATT) less than Enjoyment (ENJ). This result shows that teachers' attitudes towards STEM teaching practices are influenced primarily by affective factors and, secondarily, by cognitive ones. It is widely recognized that attitudes represent the cognitive and affective approach to influencing behaviors [162, 163]. Of interest to educational authorities is to know which are the most important predictors of attitudes and the order of their influence. In the present study, the most important predictor of teachers' attitudes is their opinion of the level of knowledge, followed by enjoyment and uselessness. A first explanation of the fact that enjoyment has a greater influence than usefulness on the attitude of teachers can be formulated, considering the fact that STEM teaching practices are one of the most emotionally challenging jobs [163]. A successful lesson involved many hours of preparation, during which the teacher analyzes several scenarios and possible developments in the conduct of the teaching activity. In addition, without the teacher's conviction that the chosen method will have a positive impact on the learning process, the method will not be implemented, however cognitively tempting it may be. From our perspective, this is the main reason why ENJ has a small ascendant in front of the USE.

## Discuss multigroup comparisons of the research model

The similarities in construct relationships for IBL and INT can be explained by teachers' personal experiences, which do not clearly distinguish between IBL and INT in terms of usefulness and enjoyment. These results are also confirmed by the teachers' answers to open questions. Briefly, the IBL and INT advantages observed by respondents are the following: the effectiveness of the learners (30% of respondents claimed this aspect in the IBL case and only 15% in the INT case), the development of thought, the presence of an innovative character, to teach a differentiated lesson, efficiently transferring knowledge to other subjects. Unlike IBL, in the INT case, teamwork's practical nature and feasibility were particularly appreciated (25% of respondents claimed this aspect in the INT case).

Even if the mediating role of USE and ENJ is demonstrated for both IBL and INT, the role of USE is more prominent in INT than in IBL. One possible explanation for this result is that more diversified skills are needed to tackle INT methods. For example, for IBL, it is sufficient for the teacher to have a very good command of the subject he/she teaches, possibly with intermediate computer skills. When talking about an integrated approach, the teacher must have advanced knowledge from several fields, for example, mathematics, natural sciences, engineering, and computer science, which the instructor can incorporate into a coherent teaching activity. For example, to use the IBL strategy to get students to study soap and the soaping reaction, it is sufficient for the teacher to have a thorough knowledge of chemistry. If, however, an approach is desired in which students are challenged to study the mechanism of feeding in humans, then knowledge from almost all areas of the natural sciences is required. Physics is needed to describe the mechanical part of feeding (mastication, advancing the food bowl into the digestive system), chemistry is needed to describe the chemical reactions that take place in the breakdown of food, and biology is needed to describe the digestive system and the functioning of its different parts.

Most research participants teach a single discipline, leading to a one-sided approach. Another aspect to be considered is the facilities available to teachers in the pre-university

environment. Even if the discipline facilities are satisfactory, INT teaching practice requires a more complex implementation in the pre-university environment than IBL High school teachers are not initially trained for teaching using INT. Only teachers' experience or training courses enhance these skills [164]. For example, Mathematics is taught as an abstract discipline, separate from other Nature Sciences or Engineering. IBL or INT strategy used in real cases [165] was difficult to apply if usefulness and enjoyment were not highlighted. Consequently, the adoption of IBL and INT in an integrated and transdisciplinary manner is a complex experience toward the individual approach of each discipline [166]. Cognitive and affective factors such as usefulness and enjoyment were necessary to support a positive attitude of teachers to accept instructional strategies in practice. It can be concluded that the transdisciplinary nature of the STEM approach requires more complex skills of teachers to adapt IBL or INT in different educational contexts.

### Contribution to research

This research has presented a mediated multigroup model to investigate the relationships between competencies and attitudes mediated by usefulness and enjoyment across secondary teachers while controlling for their specialization. For these purposes, twenty-six item scales of COM, USE, ENJ, and ATT were self-developed or adopted from literature. The scales revealed acceptable reliability, dimensionality, and validity levels through IBM and INT samples. In theory, this study has contributed to the literature by synthesizing WST, TAM, and FLT models.

This descriptive design has some distinct benefits over similar previous studies. First, some previous studies have analyzed the influence of teachers' attitudes toward the effectiveness of teaching practices [9, 81], which are correlated with the quality of classroom teaching [167]. Nevertheless, attitudes toward teaching practice style have influenced students' attitudes regarding learning [168]. However, there are data that teachers' self-assessed technology-related skills instead of attitudes affect educational practices [18]. Alternatively, enjoyment in learning has an adaptive and indirect role [169]. Second, numerous studies have investigated the teachers' perceived usefulness as predictive of their attitudes toward digital technology [170]. The testing of mediation role of the enjoyment and usefulness between the still and will was innovative, and therefore it is justified. Third, other studies [171–173] highlighted the potential role of teaching strategy in learning and academic achievement.

### Limitations and suggestions for future research

The study presented some limitations common to all survey research: the respondent's report of their beliefs or attitudes can generate a possible bias [174]. The four scales measure a specific conceptualization. Some researchers have suggested a multidimensional approach in such contexts [175]. Because only practice can provide complete answers to our research questions, new dimensions related to skill, will, and tools are required. This approach will offer proper awareness regarding strategy style motivation and effectiveness. Not unexpectedly, the specialization of teachers of the whole sample is not balanced (see Table 1). First, this is because teachers from each discipline use the IBL and INT methods differently. STEM is a complex tool for effective teaching and depends on academic discipline [149]. Also, teachers' beliefs can influence the acceptance of IBL and INT [148]. Each teacher translated IBL and INT teaching practices into classroom activities [176] adapted to discipline context, shaping particular instruction [177]. In this study, the competencies of all respondents fall into one field, that of natural sciences. From the personal experience perspective, physics teachers are more interested in using these methods because the new national physics curricula particularly encourage STEM teaching. The quantitative imbalances observed in teachers' specialization highlight

teachers' ability to use STEM practices for different subjects. Studies focused on the relationship between teachers' discipline and IBL and INT teaching practices are relatively new, marginal research fields with few experimental studies, particular between math and science [178]. Therefore, until now, information about teacher specialization impact on STEM practices beliefs is not enough. Because it is important to hold all variables constant, in the present study, teacher specialization was used as a control variable. However, the further research on the relationships between teacher specialization and their attitude to STEM practice is therefore required.

## Conclusions

The present study examined the relationship between teachers' competencies and attitude under the influence of perceived usefulness and enjoyment for key dimensions of STEM teaching practice. This study demonstrated that perceived Usefulness partially mediated the association between Competencies and Attitude in both cases of IBL and INT. Also, the Enjoyment partially mediated the association between Competencies and Attitude in cases of IBL and INT. According to the data obtained, the influence of teaching practice style as moderator is insignificant for all direct research relationships. The similarities are interesting and open new research directions to identify the differences between the two STEM teaching practices. Obviously, the relationship between competence and attitude is direct and intense, and usefulness and enjoyment have an important and similar mediating role for IBL and INT. Even if the direct effect of the COM factor on the USE, ATT and ENJ is strong for IBL and INT, the direct effect of USE and ENJ on ATT is smaller. Understanding this particular relation could improve the performance of integrated STEM education. These results reinforce the belief that competencies primarily influence attitude and not the teacher's perceived benefits in the short term.

The integrated character of STEM addresses transcending relationships between distinct disciplines, considering the consequences of the circulation of information among different branches of knowledge, allowing unity in diversity. Transdisciplinary aims to reveal the nature and characteristics of the flow of educational information. Consequently, a new language that facilitates the dialogue between specialists in different fields was created. Such a vision also makes it more challenging to transfer STEM knowledge to the application level using IBL or INT. This process involves a broader approach than a single discipline's perspective.

Another factor influencing the perception of the usefulness of IBM and INT and its relationship with teachers' attitudes is the continuous decrease of hours allocated to natural disciplines in favor of humanist disciplines. In a context where science and technology are constantly evolving, STEM teachers no longer have time to go through the specific content of their teaching fields. They have less time for integrated and transdisciplinary approaches to transfer particular competencies, as IBL and INT teaching practices separately or together ensured. Future research on how the particularities of IBL and INT can affect students' academic results were recommended. Another topic of interest for the following studies is how the educational climate, including teaching time, influences the quality of private education in real disciplines. This study's results encourage continued promotion of the IBL and INT teaching practices to teach subjects in a cross-disciplinary and integrated manner using the investigation approach. Understanding the causal relations between competencies and attitude could bring helpful insights into the teachers' STEM experiences.

## Supporting information

**S1 Table. Survey questionnaire.**
(DOCX)

**S2 Table. Raw database.**
(CSV)

**S3 Table. Results of the MICOM procedure.**
(XLSX)

## Author Contributions

**Conceptualization:** Fabiola Sanda Chiriacescu, Bogdan Chiriacescu, Cristina Miron, Ion Ovidiu Panisoara, Iuliana Mihaela Lazar.

**Data curation:** Fabiola Sanda Chiriacescu, Bogdan Chiriacescu, Alina Elena Grecu, Cristina Miron, Ion Ovidiu Panisoara, Iuliana Mihaela Lazar.

**Formal analysis:** Fabiola Sanda Chiriacescu, Bogdan Chiriacescu, Alina Elena Grecu, Cristina Miron, Ion Ovidiu Panisoara, Iuliana Mihaela Lazar.

**Investigation:** Fabiola Sanda Chiriacescu, Bogdan Chiriacescu, Alina Elena Grecu, Cristina Miron, Ion Ovidiu Panisoara, Iuliana Mihaela Lazar.

**Methodology:** Fabiola Sanda Chiriacescu, Bogdan Chiriacescu, Alina Elena Grecu, Cristina Miron, Ion Ovidiu Panisoara, Iuliana Mihaela Lazar.

**Supervision:** Fabiola Sanda Chiriacescu, Bogdan Chiriacescu, Alina Elena Grecu, Cristina Miron, Ion Ovidiu Panisoara, Iuliana Mihaela Lazar.

**Validation:** Fabiola Sanda Chiriacescu, Bogdan Chiriacescu, Alina Elena Grecu, Cristina Miron, Ion Ovidiu Panisoara, Iuliana Mihaela Lazar.

**Writing – original draft:** Fabiola Sanda Chiriacescu, Bogdan Chiriacescu, Alina Elena Grecu, Cristina Miron, Ion Ovidiu Panisoara, Iuliana Mihaela Lazar.

**Writing – review & editing:** Fabiola Sanda Chiriacescu, Bogdan Chiriacescu, Alina Elena Grecu, Cristina Miron, Ion Ovidiu Panisoara, Iuliana Mihaela Lazar.

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
