## [Decision Letter · Decision Letter 0]

20 Jul 2021

PONE-D-21-14168

Secondary teachers' competencies and attitude: A moderated mediated model based on usefulness, enjoyment and instructional strategy style

PLOS ONE

Dear Dr. Lazar,

Thank you for submitting your manuscript to PLOS ONE. After careful consideration, we feel that it has merit but does not fully meet PLOS ONE’s publication criteria as it currently stands. Therefore, we invite you to submit a revised version of the manuscript that addresses the points raised during the review process.

We look forward to receiving your revised manuscript.

Kind regards,

Manuel Santos-Trigo, PhD

Academic Editor

PLOS ONE

Additional Editor Comments:

There are three related parts that authors need to revise in order to clarify the aim of the study, the review and analysis of related studies to support coherently the research, and the pertinence of using a Likert scale questionnaire to assess participants' competences, usefulness, enjoyment, attitude regarding their teaching practices that involve IBL & STEM.

1. There is a conceptual confusion in this study when authors intend to contrast an instructional approach (IBL) that fosters students’ inquisitive method to understand concepts and to solve problems with a STEM perspective to integrate four domains (Science, Technology, Engineering, Mathematics) to direct and support students’ education. That is, an IBL approach can be used to frame teaching and learning scenarios that aim to understand concepts and to solve problems that demand the use of knowledge and strategies from the four disciplines. They are not separate domains or areas to contrast, but one (IBL) is an instructional approach based on questions that learner pose and pursue to develop disciplinary knowledge and strategies to solve problems. That is, IBL often is used to frame learning scenarios that promote students' learning and integration STEM disciplines

2. In the literature review, it is important to discuss the extent to which both IBL teaching approaches have been implemented in learning the four disciplines (Science, Technology, Engineering, and mathematics) and also studies that address the integration of these four disciplines as an educational model in terms of the main parameters or variables that are investigated in this study.

3. It is clear that the use of a Likert scale questionnaire offers general information regarding participants' consistent teaching behaviors that reflect their actual teaching competencies or even attitude or enjoyment associated with their teaching practices. In my view, the information provided by the participants' response to the questionnaire needs to be supported and constructed via, for example, the use interviews in which a reduced sample of participants could elaborate on their questionnaire answer. That is, a research blending design that incorporates both quantitative (the questionnaire) and a qualitative approach is important to enhance the trustwordiness of the study.

Journal Requirements:

Reviewers' comments:

Reviewer's Responses to Questions

**Comments to the Author**

1. Is the manuscript technically sound, and do the data support the conclusions?

Reviewer #1: No

Reviewer #2: Yes

2. Has the statistical analysis been performed appropriately and rigorously? 

Reviewer #1: No

Reviewer #2: Yes

3. Have the authors made all data underlying the findings in their manuscript fully available?

Reviewer #1: No

Reviewer #2: Yes

4. Is the manuscript presented in an intelligible fashion and written in standard English?

Reviewer #1: Yes

Reviewer #2: Yes

5. Review Comments to the Author

Reviewer #1: 1. What are the novelty of this study?

2. Literature review need more to elaborate relationship between the variables measures

3. Methodology should be clear,

- Sample involved in pilot study and actual study

- sampling method

- population number

4. find citation for result RMSEA more than 0.08? refer competences, enjoyment and attitude?

5. Show the result of AVE and composite reliability for each variables.

6. X2/df more than 0.5, refer to enjoyment and attitude, find citation to accept this value?

7. Why this study use Amos to analyze the data? all assumption of AMOS should be achieved.

8. Based on descriptive statistics values shows data was not normal distribution, refer skewness and kurtosis values, why this study continue using Amos?

9. Correlation between variable too high (more than 0.85). Meaning data has problem with multicollinearity issues.

10. Finding elaboration use word of moderator. but in literature mediator. its was difference terms and function.

11. Discussion not relate to the problem statement

12. Overall, this manuscript should be re-analysis and re-write. Please refer all the assumption using Amos software and do the step by step correctly.

Reviewer #2: This study is coherent and rigorous, both in terms of theoretical underpinning and methodology. It adequately sets out the research objectives and study hypothesis. The data analysis processes are robust and in line with the research objectives. Results are well described and respond to the research questions. Conclusions are appropriate. Authors should review the bibliography, as they do not use the "Vancouver" style, as indicated in the journal's guidelines.

6. PLOS authors have the option to publish the peer review history of their article (what does this mean?). If published, this will include your full peer review and any attached files.

Reviewer #1: No

Reviewer #2: No

---

## [Author Response · Author response to Decision Letter 0]

2 Dec 2021

Dear Additional Editor and reviewers, 

Sincerest thanks to Editor and reviewers for their valuable time and useful contribution.

The authors hope that a revised version of the manuscript will still be considered by PLoS ONE journal. 

As suggested by the reviewers, the corrections have been made. The authors corrected the manuscript in the change mode ('track changes'). The corrected version of the article indicates where the changes were made following the reviewer suggestions. The authors present an improved version of the manuscript. There are minor differences between 'track changes' and the final version due to the paper format and English language corrections. 

We incorporated all reviewer’s suggestions, and we are grateful to reviewers for careful reading of the paper and helpful suggestions and comments to improve our work. 

We provide our responses point by point and modify the manuscript according to the accurate and intuitive reviewers’ comments and additional Editor comments. 

Additional Editor Comments:

There are three related parts that authors need to revise in order to clarify the aim of the study, the review and analysis of related studies to support coherently the research, and the pertinence of using a Likert scale questionnaire to assess participants' competences, usefulness, enjoyment, attitude regarding their teaching practices that involve IBL & STEM.

1. There is a conceptual confusion in this study when authors intend to contrast an instructional approach (IBL) that fosters students’ inquisitive method to understand concepts and to solve problems with a STEM perspective to integrate four domains (Science, Technology, Engineering, Mathematics) to direct and support students’ education. That is, an IBL approach can be used to frame teaching and learning scenarios that aim to understand concepts and to solve problems that demand the use of knowledge and strategies from the four disciplines. They are not separate domains or areas to contrast, but one (IBL) is an instructional approach based on questions that learner pose and pursue to develop disciplinary knowledge and strategies to solve problems. That is, IBL often is used to frame learning scenarios that promote students' learning and integration STEM disciplines.

R: Based on supplementary searching we reorganised and extended the theoretical part of this study to assume adequate contribution to the literature (See Abstract line 17-39, Introduction line 46-58, manuscript version with ”track changes”). 

2. In the literature review, it is important to discuss the extent to which both IBL teaching approaches have been implemented in learning the four disciplines (Science, Technology, Engineering, and mathematics) and also studies that address the integration of these four disciplines as an educational model in terms of the main parameters or variables that are investigated in this study. 

R: The contribution of this study comes from the theoretical research model tested for two type pf STEM teaching practices: Integration of STEM content (INT) and Inquiry-based learning (IBL) [1]. Accordingly with an extensive literature review, we re-write all manuscript. 

3. It is clear that the use of a Likert scale questionnaire offers general information regarding participants' consistent teaching behaviors that reflect their actual teaching competencies or even attitude or enjoyment associated with their teaching practices. In my view, the information provided by the participants' response to the questionnaire needs to be supported and constructed via, for example, the use interviews in which a reduced sample of participants could elaborate on their questionnaire answer. That is, a research blending design that incorporates both quantitative (the questionnaire) and a qualitative approach is important to enhance the trustwordiness of the study.

R: The results of an interview were described in the Discussion part.

......

Dear Reviewer 1, 

Sincerest thanks for your valuable time and useful contribution.

We incorporated all your suggestions, and we are grateful to you for careful reading of the paper and helpful suggestions and comments to improve our work. 

We provide our responses point by point and modify the manuscript according to the accurate and intuitive comments.

1. What are the novelty of this study?

R: The novelty of the study was completed as the result of supplemental searching. It involves a systematic search for studies considered relevant to the topic of research. The contribution of this study comes from the theoretical research model tested for two type pf STEM teaching practices: Integration of STEM content (INT) and Inquiry-based learning (IBL) [1]. 

2. Literature review need more to elaborate relationship between the variables measures

R: Based on supplementary searching we reorganised and extended the theoretical part of this study to assume adequate contribution to the literature (See Abstract line 17-39, Introduction line 46-58, manuscript version with ”track changes”).

3. Methodology should be clear

- Sample involved in pilot study and actual study

- sampling method

- population number

find citation for result RMSEA more than 0.08? refer competences, enjoyment and attitude?

Show the result of AVE and composite reliability for each variables.

X2/df more than 0.5, refer to enjoyment and attitude, find citation to accept this value?

Why this study use Amos to analyze the data? all assumption of AMOS should be achieved.

Based on descriptive statistics values shows data was not normal distribution, refer skewness and kurtosis values, why this study continue using Amos?

Correlation between variable too high (more than 0.85). Meaning data has problem with multicollinearity issues.

Finding elaboration use word of moderator. but in literature mediator. its was difference terms and function.

R: The authors re-analysis and re-write the methodology part (See Abstract line 322-475, manuscript version with ”track changes”). 

Discussion not relate to the problem statement

R: The authors reviewed the discussion (See Discussion line 596-655, manuscript version with ”track changes”)

Overall, this manuscript should be re-analysis and re-write. Please refer all the assumption using Amos software and do the step by step correctly.

R: The authors used the SmartPLS software instead of Amos 

.....

Dear Reviewer 2, 

Sincerest thanks for your valuable time and useful contribution.

We incorporated all your suggestions, and we are grateful to you for careful reading of the paper and helpful suggestions and comments to improve our work. 

We provide our responses point by point and modify the manuscript according to the accurate and intuitive comments.

Reviewer #2: 

This study is coherent and rigorous, both in terms of theoretical underpinning and methodology. It adequately sets out the research objectives and study hypothesis. The data analysis processes are robust and in line with the research objectives. Results are well described and respond to the research questions. Conclusions are appropriate. 

Authors should review the bibliography, as they do not use the "Vancouver" style, as indicated in the journal's guidelines.

R: The authors agree that were omissions in the bibliography. Consequently, the bibliography was improved.

We hope you find this revision appropriate for research topic. Once more, thank you for your exhaustive review.

Sincerely, 

Dr. Iulia Lazar, corresponding author

---

## [Decision Letter · Decision Letter 1]

17 Jan 2022

PONE-D-21-14168R1Secondary teachers' competencies and attitude: A mediated multigroup model based on usefulness and enjoyment to examine the differences between key dimensions of STEM teaching practicePLOS ONE

Dear Dr. Lazar,

Thank you for submitting your manuscript to PLOS ONE. After careful consideration, we feel that it has merit but does not fully meet PLOS ONE’s publication criteria as it currently stands. Therefore, we invite you to submit a revised version of the manuscript that addresses the points raised during the review process.

We look forward to receiving your revised manuscript.

Kind regards,

Haoran Xie

Academic Editor

PLOS ONE

Additional Editor Comments:

Reviewer #1:

1. Three hundred teachers (i.e., mathematics, physics, chemistry, biology, computing, ICT, 330 Technologies) who used IBL and INT teaching practices have been interviewed. How to make sure 300 teachers are interviewed? How to be conducted?

2. How to get the number of 300 sample from the population? What sampling method used?

3. Based on demographic, the specialization of teachers are not balances. How can this study concluded that all the teachers can be generalized to the population number based on specialization?

4. Show the results of effect size

5. Discussion should be referred to the finding. Discuss the significant indirect effects for total sample (a), IBL (b) and INT (c) teaching practices. Discuss with pro and contra. Add citations to support the statement.

7. PLOS authors have the option to publish the peer review history of their article (what does this mean?). If published, this will include your full peer review and any attached files.

Reviewers' comments:

Reviewer's Responses to Questions

**Comments to the Author**

1. If the authors have adequately addressed your comments raised in a previous round of review and you feel that this manuscript is now acceptable for publication, you may indicate that here to bypass the “Comments to the Author” section, enter your conflict of interest statement in the “Confidential to Editor” section, and submit your "Accept" recommendation.

Reviewer #1: (No Response)

2. Is the manuscript technically sound, and do the data support the conclusions?

Reviewer #1: Partly

3. Has the statistical analysis been performed appropriately and rigorously? 

Reviewer #1: No

4. Have the authors made all data underlying the findings in their manuscript fully available?

Reviewer #1: Yes

5. Is the manuscript presented in an intelligible fashion and written in standard English?

Reviewer #1: Yes

6. Review Comments to the Author

Reviewer #1: 1. Three hundred teachers (i.e., mathematics, physics, chemistry, biology, computing, ICT, 330 Technologies) who used IBL and INT teaching practices have been interviewed. How to make sure 300 teachers are interviewed? How to be conducted?

2. How to get the number of 300 sample from the population? What sampling method used?

3. Based on demographic, the specialization of teachers are not balances. How can this study concluded that all the teachers can be generalized to the population number based on specialization?

4. Show the results of effect size

5. Discussion should be referred to the finding. Discuss the significant indirect effects for total sample (a), IBL (b) and INT (c) teaching practices. Discuss with pro and contra. Add citations to support the statement.

7. PLOS authors have the option to publish the peer review history of their article (what does this mean?). If published, this will include your full peer review and any attached files.

Reviewer #1: No

---

## [Author Response · Author response to Decision Letter 1]

17 Feb 2022

Dear Reviewer, 

Sincerest thanks for your valuable time and valuable contribution.

We incorporated all your suggestions, and we are grateful to you for your careful reading of the paper and helpful advice and comments to improve our work. 

We provide our responses point by point and modify the manuscript according to accurate and intuitive comments.

Due to significant changes in some parts of the manuscript, some errors cannot be corrected in the change mode (‘track changes’). In addition, we mention that the lines indicated in the responses to the reviewer correspond to the corrected final version without track changes. However, some differences can be observed between the “clean” and ~track changes~form.

All authors agree with the content of the manuscript after the second revision. Also, I confirm that co-authors all had an active part in the final manuscript.

As the corresponding author, I take responsibility for informing co-authors on time of editorial decisions, received reviews, changes made in response to the editorial review, and the content of revisions.

In addition, we mention that the lines indicated in the responses to the reviewer correspond to the corrected final version without track changes.

Reviewer #1:

1. Three hundred teachers (i.e., mathematics, physics, chemistry, biology, computing, ICT, 330 Technologies) who used IBL and INT teaching practices have been interviewed. How to make sure 300 teachers are interviewed? How to be conducted?

2. How to get the number of 300 sample from the population? What sampling method used?

R to Question 1 and Question 2: The participants and data collection procedure section has been rewritten (see Lines 325-369).

3. Based on demographic, the specialization of teachers are not balances. How can this study concluded that all the teachers can be generalized to the population number based on specialization?

R: Based on supplementary searching, we reorganized and extended the Limitations and suggestions for future research section (see Lines 588-602).

4. Show the results of effect size

R: The effect size results were included (see Lines 489-500). 

5. Discussion should be referred to the finding. Discuss the significant indirect effects for total sample (a), IBL (b) and INT (c) teaching practices. Discuss with pro and contra. Add citations to support the statement.

R: We extended the discussion part of this study (see Lines 547-555)

7. PLOS authors have the option to publish the peer review history of their article (what does this mean?). If published, this will include your full peer review and any attached files.

R: Yes, we agree with the option to publish the peer review history of their article.

Reviewers’ comments:

3. Has the statistical analysis been performed appropriately and rigorously? 

Reviewer #1: No

R: Measurement invariance composite (MICOM) results based on three steps (configural invariance, compositional invariance, & equality of composite mean values and variances) revealed that full measurement invariance has been established. This analysis was only briefly detailed because of the complexity and the already considerable size of the article. However, supplementary material was presented. (see Lines 438-446). 

Sincerely, 

Dr. Iulia Lazar, corresponding author

---

## [Decision Letter · Decision Letter 2]

27 Apr 2022

PONE-D-21-14168R2Secondary teachers' competencies and attitude: A mediated multigroup model based on usefulness and enjoyment to examine the differences between key dimensions of STEM teaching practicePLOS ONE

Dear Dr. Lazar,

Thank you for submitting your manuscript to PLOS ONE. After careful consideration, we feel that it has merit but does not fully meet PLOS ONE’s publication criteria as it currently stands. Therefore, we invite you to submit a revised version of the manuscript that addresses the points raised during the review process.

We look forward to receiving your revised manuscript.

Kind regards,

Haoran Xie

Academic Editor

PLOS ONE

Reviewers' comments:

Reviewer's Responses to Questions

**Comments to the Author**

1. If the authors have adequately addressed your comments raised in a previous round of review and you feel that this manuscript is now acceptable for publication, you may indicate that here to bypass the “Comments to the Author” section, enter your conflict of interest statement in the “Confidential to Editor” section, and submit your "Accept" recommendation.

Reviewer #1: All comments have been addressed

2. Is the manuscript technically sound, and do the data support the conclusions?

Reviewer #1: No

3. Has the statistical analysis been performed appropriately and rigorously? 

Reviewer #1: No

4. Have the authors made all data underlying the findings in their manuscript fully available?

Reviewer #1: No

5. Is the manuscript presented in an intelligible fashion and written in standard English?

Reviewer #1: Yes

6. Review Comments to the Author

Reviewer #1: 1. Method :

- Elaborate research design

- Why using snowball, it was for qualitative research

- Sub-topic on method should be include : Research design, participant, measurement, data analysis.

- EFA test result should be elaborate in part of method

- EFA result should be elaborate KMO and berlette, Communalities, eigenvalue, % cumulative of variance, rotation matrics. Make sure all of this ellaborate.

2. Results

- Elaborate based hypothesis listed

- Elaborate the assumption using SEM by smartPLS: normality test, multicollinearity, discriminant etc

3. Discussion

- Discuss the finding based hypothesis listed

- Use idea, justification and reason to discuss research finding

- elaborate the contribution of this study

7. PLOS authors have the option to publish the peer review history of their article (what does this mean?). If published, this will include your full peer review and any attached files.

Reviewer #1: No

---

## [Author Response · Author response to Decision Letter 2]

18 Jun 2022

Dear Editor,

Dear Reviewer, 

Sincerest thanks for your valuable time and valuable contribution.

We incorporated all your suggestions, and we are grateful to you for your careful reading of the paper and helpful advice and comments to improve our work. 

We provide our responses point by point and modify the manuscript according to accurate and intuitive comments.

Due to significant changes in some parts of the manuscript, some errors cannot be corrected in the change mode (‘track changes’). In addition, we mention that the lines indicated in the responses to the reviewer correspond to the corrected final version without track changes. However, some differences can be observed between the “clean” and ~track changes~ forms.

All authors agree with the content of the manuscript after the second revision. Also, I confirm that co-authors all had an active part in the final manuscript.

As the corresponding author, I take responsibility for informing co-authors on time of editorial decisions, received reviews, changes made in response to the editorial review, and the content of revisions.

In addition, we mention that the lines indicated in the responses to the reviewer correspond to the corrected final version without track changes.

Reviewer #1:

1. Method :

- Elaborate research design

- Why using snowball, it was for qualitative research

- Sub-topic on method should be include : 

o Research design, 

o Participant, 

o Measurement, 

o Data analysis,

o EFA test result should be elaborate in part of method (KMO and berlette, Communalities, eigenvalue, % cumulative of variance, rotation matrics)

- EFA result should be elaborate. Make sure all of this ellaborate.

R to Question 1 and Question 2: The method section has been partially rewritten (see the manuscript version with track changes).

3. Results

- Elaborate based hypothesis listed

- Elaborate the assumption using SEM by smartPLS: normality test, multicollinearity, discriminant etc

R: We reorganized and extended the result section (see the manuscript version with track changes).

4. Discussion

- Discuss the finding based hypothesis listed

- Use idea, justification and reason to discuss research finding

- elaborate the contribution of this study.

R: We reorganized and extended the discussion section (see the manuscript version with track changes).

We would also like to express our gratitude for all your consistent suggestions, in the absence of which we wouldn’t have succeeded in rigorously partially rewriting the manuscript. 

Sincerely, 

Dr. Iulia Lazar, corresponding author

---

## [Decision Letter · Decision Letter 3]

5 Aug 2022

PONE-D-21-14168R3Secondary teachers' competencies and attitude: A mediated multigroup model based on usefulness and enjoyment to examine the differences between key dimensions of STEM teaching practicePLOS ONE

Dear Dr. Lazar,

Thank you for submitting your manuscript to PLOS ONE. After careful consideration, we feel that it has merit but does not fully meet PLOS ONE’s publication criteria as it currently stands. Therefore, we invite you to submit a revised version of the manuscript that addresses the points raised during the review process.

Based on the review results, a Minor Revision has been suggested in the paper before making any final decision. Please revise the manuscript based on reviewers' comments accordingly. 

We look forward to receiving your revised manuscript.

Kind regards,

Haoran Xie

Academic Editor

PLOS ONE

Journal Requirements:

Reviewers' comments:

Reviewer's Responses to Questions

**Comments to the Author**

1. If the authors have adequately addressed your comments raised in a previous round of review and you feel that this manuscript is now acceptable for publication, you may indicate that here to bypass the “Comments to the Author” section, enter your conflict of interest statement in the “Confidential to Editor” section, and submit your "Accept" recommendation.

Reviewer #1: (No Response)

2. Is the manuscript technically sound, and do the data support the conclusions?

Reviewer #1: Partly

3. Has the statistical analysis been performed appropriately and rigorously? 

Reviewer #1: No

4. Have the authors made all data underlying the findings in their manuscript fully available?

Reviewer #1: No

5. Is the manuscript presented in an intelligible fashion and written in standard English?

Reviewer #1: Yes

6. Review Comments to the Author

Reviewer #1: 1. Find references for communalities value less than 0.2 are accepted

2. Find citations for percentages of total variance less than 60% are accepted.

3. Based on experience to use EFA, this study's result was not good. Some of the items should be removed. but the author still proceeds with the item with a communalities value less than 0.4 or 0.5.

4. Results pilot test for CFA are not clear. Need to show the value of factor loading, correlation, AVE, Composite reliability, and Cronbach Alpha, Not only of model fit value.

5. Actual study result need to be re-analysis after checking a pilot test result.

7. PLOS authors have the option to publish the peer review history of their article (what does this mean?). If published, this will include your full peer review and any attached files.

Reviewer #1: No

---

## [Author Response · Author response to Decision Letter 3]

2 Sep 2022

Dear Reviewer, 

All authors agree with the content of the manuscript after the fourth revision. Also, I confirm that co-authors all had an active part in the final manuscript.

As the corresponding author, I take responsibility for informing co-authors on the time of editorial decisions, received reviews, changes made in response to the editorial review, and the content of revisions.

In addition, we mention that the lines indicated in the responses to the reviewer correspond to the corrected final version without track changes.

Reviewer #1:

1. Find references for communalities value less than 0.2 are accepted

R to Suggestion 1: The Data analysis subsection has been partially rewritten after statistical corrections. Now, all communalities are larger than 0.4. See in the document “Revised 4 Manuscript”- Table 2. Factor Loadings from Exploratory Factor Analysis for total sample (N=600)

2. Find citations for percentages of total variance less than 60% are accepted.

R to Suggestion 2: Now, the percentages of total variance are higher than 60%. See in the document “Revised 4 Manuscript” – lines 474-482

3. Based on experience to use EFA, this study's result was not good. Some of the items should be removed. but the author still proceeds with the item with a communalities value less than 0.4 or 0.5.

R to Suggestion 3: The EFA analyses were revised. See in the document “Revised 4 Manuscript”– lines 474-487

4. Results pilot test for CFA are not clear. Need to show the value of factor loading, correlation, AVE, Composite reliability, and Cronbach Alpha, Not only of model fit value.

R to Suggestion 4: The CFA analyses were revised. See in the document “Revised 4 Manuscript”– lines 544-565

5. Actual study results need to be re-analysis after checking a pilot test result.

R to Suggestion 5: Actual study results were re-analysis after checking a pilot test result.

We would also like to express our gratitude for all your consistent suggestions, in the absence of which we wouldn’t have succeeded in rigorously partially rewriting the manuscript. 

Sincerely, 

Dr. Iulia Lazar, corresponding author

---

## [Decision Letter · Decision Letter 4]

8 Nov 2022

PONE-D-21-14168R4Secondary teachers' competencies and attitude: A mediated multigroup model based on usefulness and enjoyment to examine the differences between key dimensions of STEM teaching practicePLOS ONE

Dear Dr. Lazar,

Thank you for submitting your manuscript to PLOS ONE. After careful consideration, we feel that it has merit but does not fully meet PLOS ONE’s publication criteria as it currently stands. Therefore, we invite you to submit a revised version of the manuscript that addresses the points raised during the review process.

I have seen that you have made substantial revisions to the manuscript based on the previous reviewer comments. As a result, the quality of manuscript has improved, and Reviewer 1 is now very satisfied with your manuscript. However, Reviewer 4 has raised additional issues. Would you please revise your manuscript once again to address the new comments? Almost there! 

We look forward to receiving your revised manuscript.

Kind regards,

Heng Luo, Ph.D.

Academic Editor

PLOS ONE

Journal Requirements:

Reviewers' comments:

Reviewer's Responses to Questions

**Comments to the Author**

1. If the authors have adequately addressed your comments raised in a previous round of review and you feel that this manuscript is now acceptable for publication, you may indicate that here to bypass the “Comments to the Author” section, enter your conflict of interest statement in the “Confidential to Editor” section, and submit your "Accept" recommendation.

Reviewer #1: All comments have been addressed

Reviewer #3: (No Response)

Reviewer #4: (No Response)

2. Is the manuscript technically sound, and do the data support the conclusions?

Reviewer #1: Yes

Reviewer #3: Yes

Reviewer #4: Yes

3. Has the statistical analysis been performed appropriately and rigorously? 

Reviewer #1: Yes

Reviewer #3: Yes

Reviewer #4: Yes

4. Have the authors made all data underlying the findings in their manuscript fully available?

Reviewer #1: Yes

Reviewer #3: Yes

Reviewer #4: Yes

5. Is the manuscript presented in an intelligible fashion and written in standard English?

Reviewer #1: Yes

Reviewer #3: Yes

Reviewer #4: Yes

6. Review Comments to the Author

Reviewer #1: Congratulation!

Good article with great analysis reports and new knowledge.

This manuscript has a good standard to publish. But need to revise the all the figures. need clearer.

Reviewer #3: (No Response)

Reviewer #4: (No Response)

7. PLOS authors have the option to publish the peer review history of their article (what does this mean?). If published, this will include your full peer review and any attached files.

Reviewer #1: No

Reviewer #3: No

Reviewer #4: No

---

## [Author Response · Author response to Decision Letter 4]

4 Dec 2022

Dear Reviewers, 

All authors agree with the content of the manuscript after the fifth revision. Also, I confirm that co-authors all had an active part in the final manuscript.

As the corresponding author, I take responsibility for informing co-authors on the time of editorial decisions, received reviews, changes made in response to the editorial review, and the content of revisions.

In addition, we mention that the lines indicated in the responses to the reviewer correspond to the corrected final version without track changes.

Reviewer 1: 

General remarks:

The overall manuscript is very focused. This research explores the mediating role of perceived usefulness and enjoyment of STEM teaching practice between secondary teachers' competencies and attitudes through structural equation model. Meanwhile, the study examines whether the relationship between model structure differs for Inquiry-based learning (IBL) and STEM content integration (INT), which is an interesting and innovative finding. 

Answer: The manuscript has been rewritten (see manuscript with Track changes) so that sentences no longer contain redundant words, arguments are logical and coherently presented.

Specific remarks:

1. p. abstracts. The last sentence contains a statement: “Understanding the mediating role of perceived usefulness and enjoyment for each STEM practice would help teachers succesfully implement STEM education.” It needs to be explained in detail in your manuscript later on. 

Answer: The statement: “Understanding the mediating role of perceived usefulness and enjoyment for each STEM practice would help teachers implement STEM education (see p.3 line 79-85 and p.36, line 710-716)

2. p.1 lines 44-45. References to STEAM definitions are best marked with the page number of the original text. 

Answer: References to STEAM definitions have been marked with the page numbers of the original texts (see p. 1 line 44 and p.5 line 90)

3. p.4 line 68. IBL emphasizes critical thinking and inquiry-based learning. However, in IBL mode, why is it important to explore the factors affecting learners' attitudes? This logic should be further elaborated. Additional, if the term IBL appears for the first time, it is recommended to explain. 

Answer: Details on the importance of exploring the factors affecting students' attitudes towards IBL (see p.4 line 68-75)

Answer: References to the IBL definition have been marked with the page numbers of the original texts (see p. 4 line 79)

5. p.5 lines 75-82, the first sentence does not have a strong logical connection with the content elaborated later. It is suggested to further demonstrate the reasons why IBL or INT method in STEM is difficult to implement. 

Answer: The reasons why IBL or INT method in STEM is difficult to implement was demonstrated (see p. 5 line 95-101)

6. p.10, the concept logic of teacher's ability and student's ability is not clear enough in the discussion of the three elements of ability, especially in lines 115-117. It is suggested to reorganize the sentences to make the logic more coherent. 

Answer: The sentences were reorganized to make the logic more coherent (see p. 5 line 142-153)

7. p.13, lines 145-146, "Moreover, usefulness mediates the relationship between ease of use and attitude" suggest adding literature citation support. 

Answer: The literature citation support was add (see p. 8 line 180)

8. p.25, lines 235-241, the logic of the argument that attitude depends on cognitive ability needs to be reorganized. What new cognitive needs does STEM teaching practice put forward for teachers' cognitive ability, which is different from other teaching environments, and there is no logical connection between the first and second

Answer: The logic of the argument that attitude depends on cognitive ability needs was reorganized (see p. 12 line 1269-278)

9. p.27, line 253, the summary of the hypothesis can be more concise. There are similar contents in the introduction and the previous literature review, which can be carefully considered. 

 Answer: The subchapter “The current study. Hypothesis summary” was revised 

10. The WST model explains the relationship between attitude and ability, but whether the other two mediating variables can be deeply explained by different model theories, for example, whether perceived usefulness and attitude can be explained by Technology Acceptance Model (TAM), and whether enjoyment can be explained by Flow Theory. 

Answer: The reviewed manuscript synthesized WST, TAM, and FLT models to describe the relationships between secondary teachers' competencies and attitudes

11. p.54, line 473, there is no Table 2A. Please check the chart and the passage carefully.

 Answer - Resolved

12. Table 3b, line 543, place parameters in italics (p). Also check the remainder of your manuscript for this. 

 Answer- Resolved

13. In the end, the Path diagram can be refined, for example by adding a topic item. Answer-resolved

We would also like to express our gratitude for all your consistent suggestions, in the absence of which we wouldn’t have succeeded in rigorously partially rewriting the manuscript. 

Sincerely, 

Dr. Iulia Lazar, corresponding author

---

## [Editor Report · Decision Letter 5]

19 Dec 2022

Secondary teachers' competencies and attitude: A mediated multigroup model based on usefulness and enjoyment to examine the differences between key dimensions of STEM teaching practice

PONE-D-21-14168R5

Dear Dr. Lazar,

We’re pleased to inform you that your manuscript has been judged scientifically suitable for publication and will be formally accepted for publication once it meets all outstanding technical requirements.

Kind regards,

Heng Luo, Ph.D.

Academic Editor

PLOS ONE
---

## [Editor Report · Acceptance letter]

27 Dec 2022

PONE-D-21-14168R5 

Secondary teachers' competencies and attitude: A mediated multigroup model based on usefulness and enjoyment to examine the differences between key dimensions of STEM teaching practice 

Dear Dr. Lazar:

I'm pleased to inform you that your manuscript has been deemed suitable for publication in PLOS ONE. Congratulations! Your manuscript is now with our production department. 

Kind regards, 

on behalf of

Dr. Heng Luo 

Academic Editor

PLOS ONE